# Establishing superfine nanofibrils for robust polyelectrolyte artificial spider silk and powerful artificial muscles

Wenqian He[1], Meilin Wang[1], Guangkai Mei[1], Shiyong Liu[1], Abdul Qadeer Khan[1], Chao Li[1], Danyang Feng[1], Zihao Su[1], Lili Bao[2], Ge Wang[1], Enzhao Liu[3], Yutian Zhu [4], Jie Bai[5], Meifang Zhu [6] ✉, Xiang Zhou [2] ✉ & Zunfeng Liu [1] ✉

Spider silk exhibits an excellent combination of high strength and toughness, which originates from the hierarchical self-assembled structure of spidroin during fiber spinning. In this work, superfine nanofibrils are established in polyelectrolyte artificial spider silk by optimizing the flexibility of polymer chains, which exhibits combination of breaking strength and toughness ranging from 1.83 GPa and 238 MJ m$^{-3}$ to 0.53 GPa and 700 MJ m$^{-3}$, respectively. This is achieved by introducing ions to control the dissociation of polymer chains and evaporation-induced self-assembly under external stress. In addition, the artificial spider silk possesses thermally-driven supercontraction ability. This work provides inspiration for the design of high-performance fiber materials.

Acting over evolutionary time scales, nature has generated astonishing nanostructures via the self-assembly of bio-macromolecules, yielding diverse tough materials with extraordinary mechanical properties[1–4]. Spider silk exhibits an excellent combination of mechanical strength and toughness, which originates from the hierarchical structure assembled by the protein peptide chains. The hierarchical structures of spider silk include nanofibrils with spiral structure formed by highly oriented peptide chains, β-sheet crystallites serving as physical cross-linking sites, hydrogen bonding to dissipate energy, and sheath-core architecture with rigid sheath and soft core[5]. By mimicking some of these structural characteristics, great achievements have been realized in preparation of artificial spider silks with high strength and toughness by employing spidroin proteins and peptides[6,7], carbon nanotube/polymer composite fibers[8,9], and hydrogel fibers[10,11].

Artificial spider silks based on non-peptide synthetic polymers are prepared by mimicking the hierarchical structure of the spider silk,

including sheath-core[12,13], spiral alignment by inserting twist[14,15], cross-linking[11,16], and combination of the above structures[10,17]. Among these methods, draw-spinning of the polymer hydrogel fibers are proven to be an effective way to prepare spider-silk-like mechanical properties, with hierarchical structures including sheath-core and spiral alignment[10], internal cross-linking and hydrogen bonding[17], buckled sheath[18], and adhesiveness[19]. Until now, to modulate the assembly of molecular chains in the hierarchical structure of the polymer artificial spider silk is still an on-going challenge to further improve the fiber mechanical properties.

The nanofibrils, especially the nanodomain size of 2–6 nm achieved by self-assembly of the polypeptide chains in natural spider silk is considered as an important origin for its excellent mechanical strength and toughness[20,21]. Similarly, the mechanical properties of a fiber depend on the self-assembled structure[17,22,23] and degree of alignment of the molecular chains[24–26]. The ordered arrangement of

[1]State Key Laboratory of Medicinal Chemical Biology, Key Laboratory of Functional Polymer Materials, Tianjin Key Laboratory of Functional Polymer Materials, College of Chemistry, Nankai University, Tianjin 300071, China. [2]Department of Science, China Pharmaceutical University, Nanjing 211198, China. [3]Tianjin Key Laboratory of Ionic-Molecular Function of Cardiovascular disease, Department of Cardiology, Tianjin Institute of Cardiology, The Second Hospital of Tianjin Medical University, Tianjin 300211, China. [4]College of Materials, Chemistry and Chemical Engineering, Hangzhou Normal University, Hangzhou 311121, China. [5]Chemical Engineering College, Inner Mongolia University of Technology, Hohhot 010051, China. [6]State Key Laboratory for Modification of Chemical Fibers and Polymer Materials, College of Materials Science and Engineering, Donghua University, Shanghai 201620, China. ✉e-mail: zhumf@dhu.edu.cn; zhouxiang@cpu.edu.cn; liuzunfeng@nankai.edu.cn

molecular chains can increase the fiber mechanical strength, and the self-assembled nanofibrils can inhibit crack propagation during deformation, thereby increasing energy dissipation[27–29]. Therefore, being able to precisely control the self-assembly of molecular chains in nanostructures is key to improving the fiber mechanical strength and toughness[30]. Although assembly of polymer chains into nanofibrils were observed in polymer materials[31,32], it is still a challenge to regulate the self-assembly of nanofibrils in the hierarchical structure of the polymer artificial spider silk to achieve excellent mechanical properties.

In this work, a robust artificial spider silk was prepared by establishing superfine nanofibrils by optimizing the molecular chain flexibility to modify the self-assembling process. An increased degree of molecular chain alignment of polyelectrolyte was achieved through an increased dissociation degree during solvent evaporation under external stress, resulting in excellent tunable mechanical properties. For example, the polyacrylic acid fiber (PAF) exhibited a combination of mechanical strength and toughness ranging from 1.83 GPa and 238 MJ m$^{-3}$ to 0.53 GPa and 700 MJ m$^{-3}$, respectively. A nanodomain size of 5.2 nm were observed in the self-assembly structure, approaching that of natural spider silk[20], which is considered as an important origin for improving the mechanical strength and toughness.

A thermally driven supercontraction behavior was observed for the PAF-based artificial spider silk. Interestingly, the establishment of superfine nanofibrils provided the obtained PAF-based artificial spider silk highly increased actuation properties. The PAF exhibited the maximum actuation stress of 65 MPa, and the maximum work capacity of 2.77 J g$^{-1}$ for the optimized dissociation degree. Further, twist insertion followed by cross-linking produced coiled artificial muscles with reversible actuation. The current work provides a design strategy for high-performance, smart fibers for use in soft robotics, flexible electronics and intelligent devices.

## Results

### Establishing nanofibrils for hierarchically structured polyelectrolyte artificial spider silk

The polypeptide chains of the spindroins exhibited high flexibility in the spinning dope of the spider, which self-assembles into fine nanofibrils during spinning into the spider silk exhibiting hierarchical structure. Here we investigated the establishment of superfine nanofibrils of polyelectrolyte artificial spider silk by optimizing the polymer chain flexibility (Fig. 1). The artificial spider silk was prepared from hydrophilic polymer electrolytes, such as poly(acrylic acid) (PAA), polyacrylamide (PAM), and poly(2-(Dimethylamino)ethyl methacrylate) (PDMAEMA). The PAF was prepared as follows. A 0.5-mm-diameter capillary tube was filled with a mixture solution for polymerization, which contained the monomer acrylic acid (40 wt%) and the initiator ammonium persulfate (0.2 wt% relative to AA). It was subjected to photo-polymerization under an ultraviolet (UV) lamp with 365-nm wavelength (20 W) for 2 h, and then the capillary tube was broken in the middle to pull out the PAF. The as-prepared wet PAF contained 55.3% of water and was highly elastic; it could return to the initial length by releasing the applied stress and could be repeatedly stretched multiple times. To obtain a dry PAF, the wet PAF was stretched to thirteen times its initial length (1200% strain) and then dried in an oven at 60 °C (Supplementary Fig. 1). The PAF retained its elongated length for a drying time longer than 5 min. The PAF exhibited a sheath-core structure under the reflection mode of the Metallographic microscope (Supplementary Fig. 2a). The fiber sheath contains less water and shows less transparent, and therefore exhibited a high modulus. The fiber core is highly transparent and contains large amount of water, and therefore exhibited a low modulus and elasticity. Such a sheath-core structure provides a rigid-yet-flexible architecture. Accompanying with the hydrogen bonding network between polymer

chains, this provides a good platform for establishing hierarchical structures for polymer artificial spider silk. With increase of the drying time from 5 to 60 min, we observed an increment of the fiber sheath and a decrement of the fiber core, with fiber sheath-ratio increased from 31 to 77%. As the drying time increased from 0.5 h to 3 h, the water content decreased monotonically from 11.7% to 5.4% and reached a plateau. The breaking strength of the PAF increased monotonically from 0.25 GPa to reach a plateau value of 0.37 GPa, and the PAF exhibited the maximum fracture strain (86%) and toughness (200 MJ m$^{-3}$) at a water content of 8.9% for a drying time of 2 h; both the fracture strain and toughness decreased with a further decrease in the water content (Supplementary Fig. 2b).

We then characterized the nanofibril structure of the PAF via water-evaporation-induced self-assembly under external stress. Microstripes with a width of 5.2 μm were observed on the PAF surface aligned along the fiber axis, and spider-silk-like nanofibrils with a width of 34 nm were observed in the longitudinal cross section for a drying time of 3 h (Fig. 1d and Supplementary Fig. 3). The PAF dried without being subjected to a pre-strain (control sample) showed a smooth surface (Supplementary Fig. 4). Under uniaxial tension, the polymer chains orients along the direction of stretching and further drying facilitates the polymer chains easily aggregated to form nanofibrils under the confined configuration through intermolecular hydrogen bonds. As the drying time increased, the nanofibrils became increasingly evident and aligned more and more closely, gradually converging to the parallel state, which manifested that the polymer chains inside the fiber became more ordered. The POM images show that the PAF exhibited increasing birefringence with increasing drying time (Supplementary Fig. 5), confirming increased polymer chain alignment along the fiber axial direction during water evaporation. After tensile failure, the nanofibrils were observed to be pulled out of the fiber cross section, which represents a typical ductile fracture behavior and contributes to the fiber toughness[33,34] (Supplementary Fig. 6).

### Modulating the nanofibrils structure and the mechanical properties by controlling the dissociation degree

The aligned nanofibrils is key for improving the fiber mechanical properties, which can effectively increase the fiber strength and inhibit crack propagation during deformation. Therefore, we attempted to modulate the size and alignment of the nanofibrils in the hierarchically structured artificial spider silk by tuning the molecular chain flexibility to enhance the mechanical properties of the fibers. The polyacrylic acid chains with a higher dissociation degree exhibited an increased binding affinity of the water molecules and therefore showed higher molecular flexibility, favoring the re-organization of the molecular chains to form nanofibrils. We further optimized the self-assembly of the PAF molecular chains by controlling the dissociation degree ($\alpha$) of the polyacrylic acid chains by adding an acid or a base. Here, PAF$_\alpha$ represents PAF with different dissociation degrees. Adding HCl during the polymerization of acrylic acid resulted in a decreased dissociation degree, while adding NaOH resulted in an increased dissociation degree. For example, $\alpha$ was 0.017% for a molar ratio of acrylic acid to HCl ($n_{AA}$:$n_{HCl}$) of 10:1, and it was 4.75% for a molar ratio of acrylic acid to NaOH ($n_{AA}$:$n_{NaOH}$) of 10:1 (Supplementary Table 3). With increasing $\alpha$ from 0.17% to 96.5%, the water content of the PAF$_\alpha$ increased from 5.4% to 13.2% (Supplementary Table 4).

The mechanical characterization of the PAF revealed that its breaking strength and toughness decreased by adding different types of acids, such as CH$_3$COOH, CH$_3$SO$_3$H, and B(OH)$_3$ (with a molar ratio of each acid to the acrylic acid of 1:8). On the other hand, the breaking strength and toughness of the PAF increased by adding different types of bases, such as LiOH, KOH, and NH$_3$·H$_2$O (with a molar ratio of each base to the acrylic acid of 1:9) (Supplementary Fig. 8, Supplementary Tables 5 and 6). We then systematically investigated the dependence of the fiber mechanical properties on the dissociation

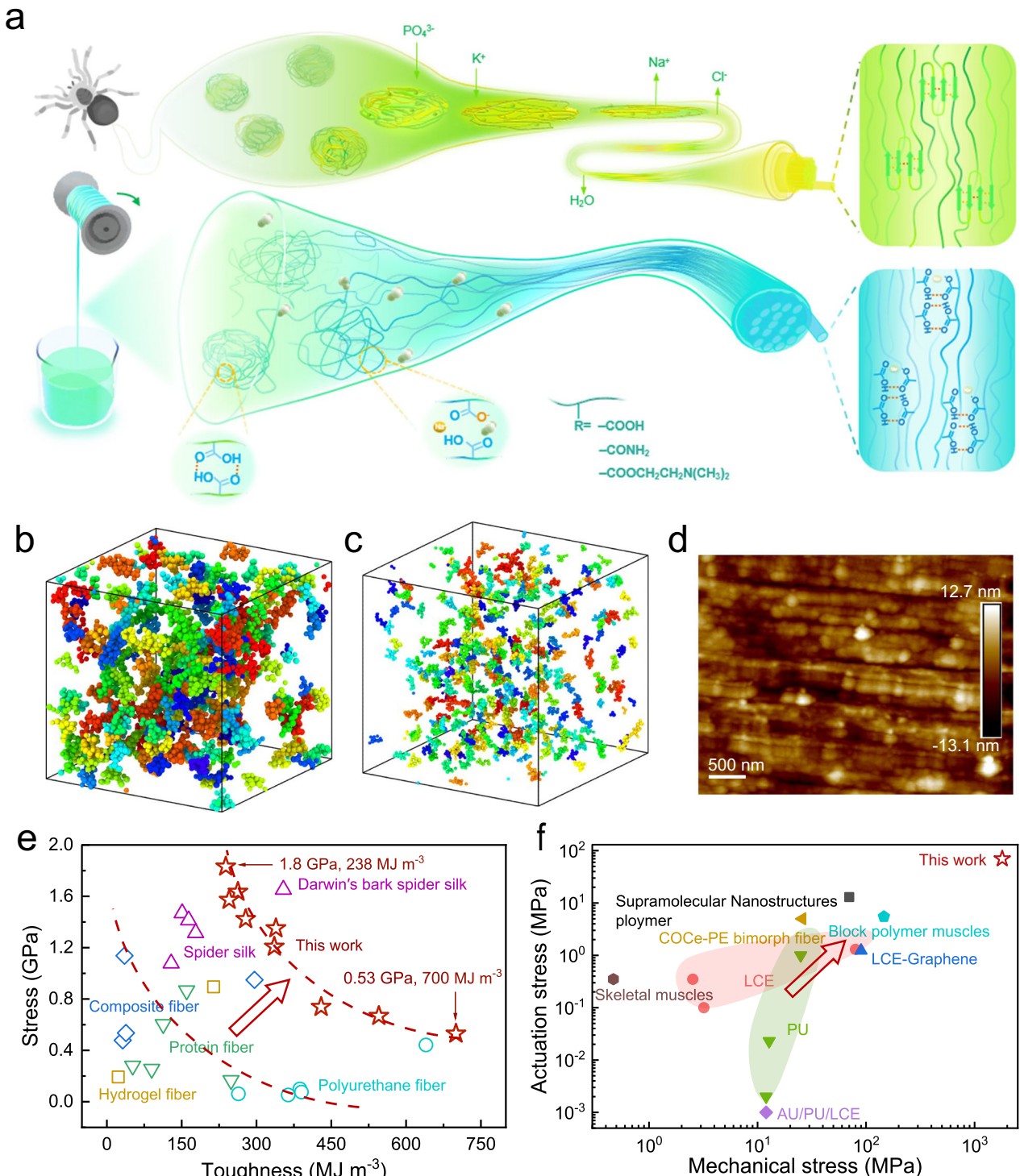

**Fig. 1 | Preparation and characterization of the PAF$_\alpha$-based artificial spider silk.** **a** Schematic of the spinning of spider silk and the PAF$_\alpha$ artificial spider silk. The modulation of the nanofibrils of the PAF$_\alpha$ achieved by tuning the polymer chain flexibility, which in turns is obtained by changing the dissociation degree. **b**, **c** The coarse-grained MD simulations show that the polymer chain clusters transform from a multi-branched shape to highly dissociated as $\alpha$ increases from 0% to 25%.

**d** AFM image of the longitudinal-sectional nanofibrils of the PAF$_{0.17\%}$. **e** Comparison of the breaking stress and toughness of the artificial spider silk fibers in this work with those of typical robust fiber materials reported in the literature. **f** Comparison of the actuation stress and mechanical strength of the drawn-spun PAF$_{1.1\%}$ prepared in this work with those of typical thermoresponsive shape memory polymers artificial muscles reported in the literature.

degree, especially by adding a base. Unless otherwise specified, NaOH was employed as the base to obtain the PAF$_\alpha$ in the following context. PAF$_\alpha$ with $\alpha$ values ranging from 1.38% to 96.5% were prepared for the molar ratio of acrylic acid to NaOH ($n_{AA}$:$n_{NaOH}$) values ranging from 20:1 to 1:1, respectively. Similarly, PAF$_\alpha$ with $\alpha$ values ranging from 0.017% to

0.044% were also prepared for the molar ratio of acrylic acid to HCl ($n_{AA}$:$n_{HCl}$) values ranging from 10:1 to 40:1, respectively. As $\alpha$ increased from 0.017% to 4.75%, the breaking strength, fracture strain, toughness, and energy dissipation increased from 0.11 GPa, 54%, 45 MJ m$^{-3}$, and 33 MJ m$^{-3}$ to the maximum values of 0.52 GPa, 110%, 390 MJ m$^{-3}$, and

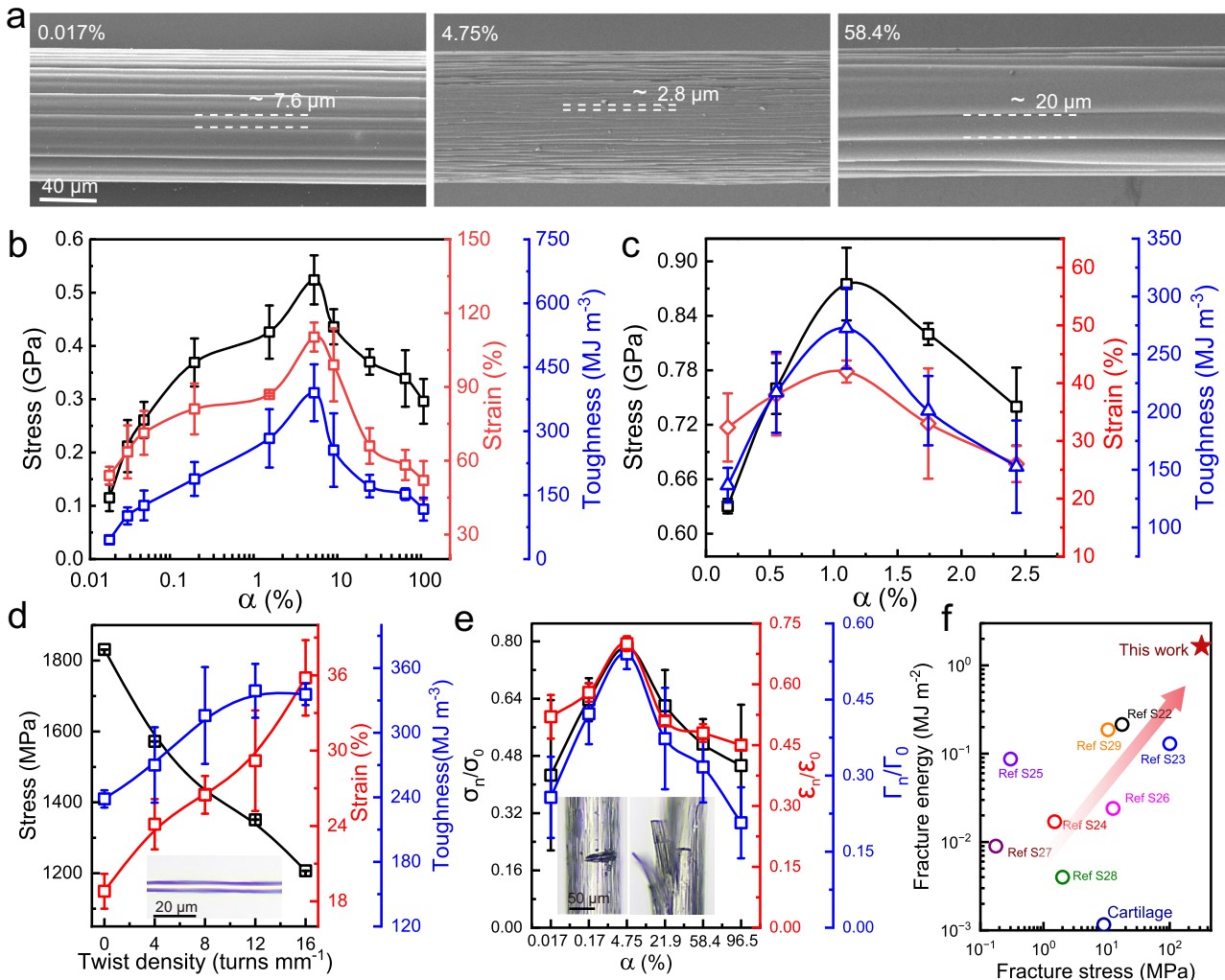

**Fig. 2 | Mechanical properties of the PAF$_\alpha$-based artificial spider silk. a** SEM images showing the evolution of the surface microstrips of the PAF under different $\alpha$ values. Breaking stress, breaking strain, and toughness of the PAF$_\alpha$ (**b**) and the draw-spun PAF$_\alpha$ (**c**). **d** Breaking stress, breaking strain, and toughness of the draw-spun PAF$_{1.1\%}$ under different twist densities. The insets in **d** show the metallographic microscope image of PAF$_{1.1\%}$ with a twist density of 8 turns mm$^{-1}$. The fiber diameter was 80 μm for **b** and 5 μm for **c** and **d**. The stretch rate in the mechanical tests was 200 mm min$^{-1}$ for **b** and 20 mm min$^{-1}$ for **c** and 500 mm min$^{-1}$ for **d**. **e** Ratio of the breaking stress, breaking strain, and fracture toughness of a PAF$_\alpha$ with a notch to those of a PAF$_\alpha$ without a notch ($\sigma_n/\sigma_0$, $\varepsilon_n/\varepsilon_0$, and $\Gamma_n/\Gamma_0$, respectively). The optical images in the inset show the notched PAF$_{4.75\%}$ before and after breakage. **f** Comparison of the fracture energy and fracture stress of the notched PAF$_{4.75\%}$ prepared in this work with those of typical tough materials reported in the literature. The error bars for **b**–**e** represent mean ±SD ($n = 5$ independent samples).

367 MJ m$^{-3}$, respectively; they then decreased as $\alpha$ increased further to 96.5% (Fig. 2b and Supplementary Figs. 9 and 10). The dependence of the mechanical properties on the dissociation degree is also applicable to other types of polyelectrolyte fibers, such as the co-polymer of AM and DMAEMA. The p(AM-co-DMAEMA) fibers are weak and fragile; thus, they can be hardly taken out of the capillary. The fibers can be taken out by adding HCl (with 1.5:100 to the monomer mass, with corresponds to $\alpha = 8.7\%$). As $\alpha$ increased from 8.7% to 48.8%, the breaking strength increased from 215 MPa to the maximum value of 364 MPa, and the fracture strain increased from 38.7% to the maximum value of 47.5%; the breaking strength and fracture strain decreased with a further increase in $\alpha$ (Supplementary Fig. 11).

We next characterized the structure and morphology of the PAF$_\alpha$ for different $\alpha$ values by employing atomic force microscopy (AFM), scanning electron microscopy (SEM), polarized optical microscopy (POM), and two-dimensional small-angle X-ray scattering (2D SAXS), 2D wide-angle X-ray scattering (WAXS), and Fourier-transform infrared (FTIR) spectroscopy (Fig. 3). The FTIR spectra indicate that the content of the COO$^-$ group (1540 cm$^{-1}$) increased and that of the COOH group (1700 cm$^{-1}$) decreased with increase of the dissociation degrees

(Supplementary Fig. 7). The POM images show that the color of the PAF$_\alpha$ changed initially from cyan to purple and then to yellow with increasing brightness as $\alpha$ increased from 0.028% to 4.75%, indicating an increased degree of molecular chain alignment[35]; as $\alpha$ increased further to 58.4%, the PAF$_\alpha$ exhibited blue color with decreasing brightness (Fig. 3a). SEM was employed to investigate the surface morphology and the nanofibrils in the longitudinal cross section of the PAF$_\alpha$ (Fig. 2a and Supplementary Fig. 12 and 13). As $\alpha$ increased from 0.017% to 4.75%, the width of the surface microstrips decreased from 7.6 to 2.8 μm; it then increased to 20 μm as $\alpha$ increased further to 58.4% and finally vanished for $\alpha$ of 96.5%. We next investigated the longitudal sectional morphology of PAF$_\alpha$ fiber with different dissociation degrees. Loose and blurry nanofibrils with ridial length larger than 133 nm were observed for $\alpha$ < 0.044%, which would be attributed to the weak aggregation of the polymer chains resulting from decreased hydrogen bonding by addition of HCl. The PAF$_\alpha$ showed aligned, denser and thinner nanofibrils with increase of $\alpha$, which exhibited a minimum nanofibril diameter of 17.7 nm for $\alpha$ of 4.75%. The nanofibril almost disappeared as $\alpha$ further increased to 58.4%, showing a nearly flat fiber longitudal section. The above structural changes of the

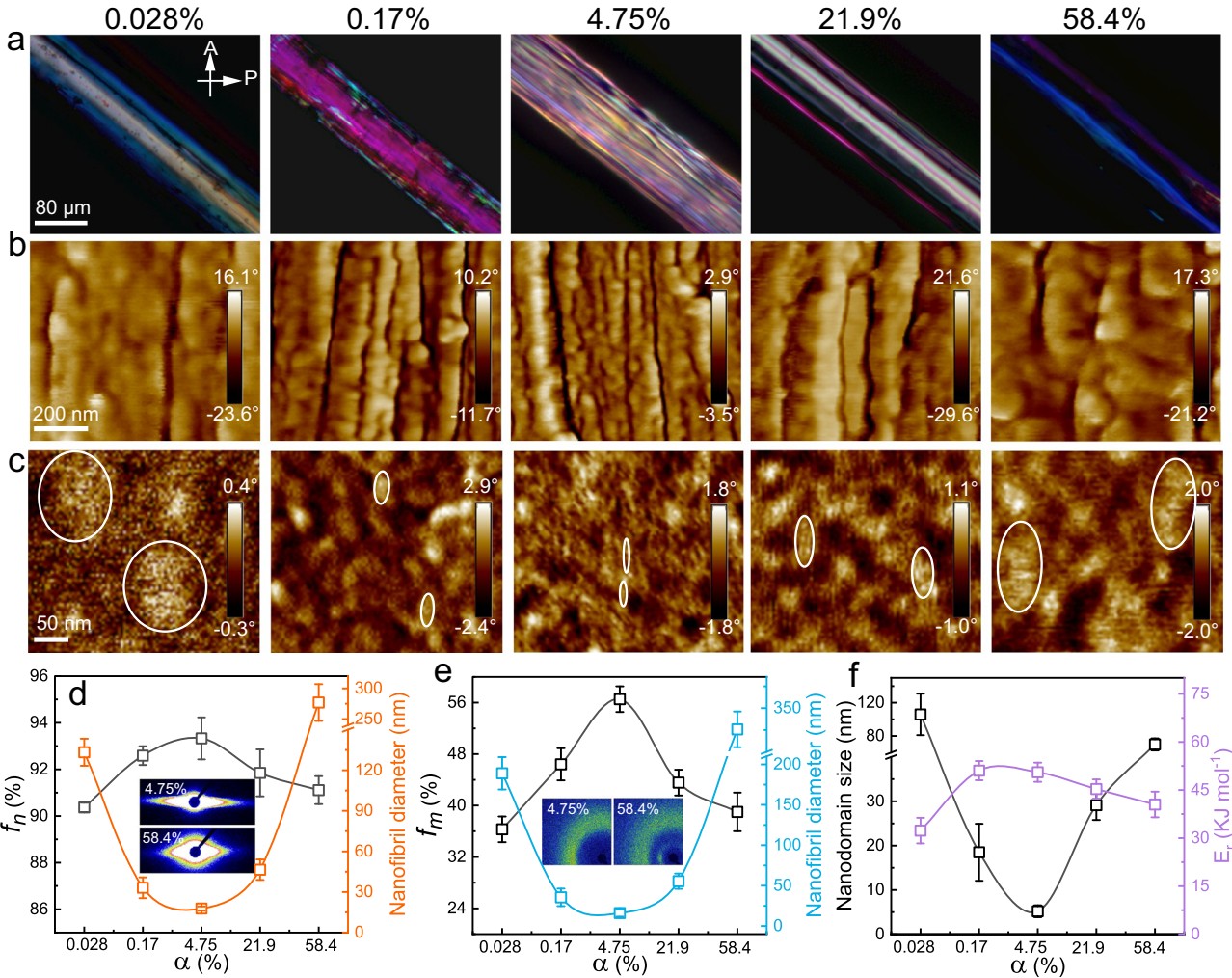

**Fig. 3 | Reinforcing mechanisms of the PAF$_\alpha$: from nanofibrils to molecular chains. a** POM image and **b** longitudinal sectional AFM phase image of the PAF$_\alpha$ with different $\alpha$ values. **c** AFM phase images of the surface of polyacrylic acid films with different $\alpha$ values that were prepared by bi-axial stretching of a polyacrylic gel. **d** Orientation degree of the nanofibrils ($f_n$) and nanofibril diameter obtained from the SEM images. **e** Orientation degree of the molecular chains ($f_m$) and nanofibril diameter obtained from the AFM images. **f** Nanodomain size obtained from the AFM phase images of the polyacrylic acid films with different $\alpha$ values and the free energy of polymer chain relaxation ($E_r$) obtained from the *Arrhenius* equation. The error bars for **d**–**f** represent mean ± SD ($n = 3$ independent samples).

nanofibrils corresponded well with the dependence of the mechanical properties on the dissociation of the polymer chains. This indicates that appropriate dissociation degree range of the PAF$_\alpha$ facilitates the formation of fine nanofibrils, which resulted in the best mechanical properties.

The phase image obtained via AFM also shows the presence of nanofibrils along the longitudinal section of the PAF$_\alpha$ (Fig. 3b). As $\alpha$ increased from 0.028% to 4.75%, the diameter of the longitudinal-sectional nanofibrils decreased from 189 to 16.1 nm, and it then increased to 316 nm as $\alpha$ increased further to 58.4% (Fig. 3e). In order to observe smaller nanodomains, we prepared flat polyacrylic acid films with different $\alpha$ values and characterized their surface using AFM (Fig. 3c and Supplementary Figs. 14 and 15). The polyacrylic acid film was obtained by biaxially stretching the bulk gel to form a film followed by air drying for 12 h while tethering on a circular frame. The nanodomains were observed for the dried film under AFM. The PAF film showed a uniform morphology with negligible nanodomain for $\alpha$ of 0.017%, and distinct nanodomains were observed as $\alpha$ increased to 0.044%. The nanodomains became more delicate with further increase in $\alpha$, and the average size decreased to a minimum average value of 5.2 nm at $\alpha$ of 4.75%, approaching to that of the nature spider silk (~4 nm)[20]. With further increase of $\alpha$, the nanodomain size dramatically

increased, and finally disappeared at $\alpha$ of 96.5%. These results further confirmed that modulating the dissociation degree of polymer chains for high flexibility can achieve delicate nanodomain structure, thus improving the mechanical properties of the artificial silk and film.

We measured the mechanical properties of the PAA films by adjusting the dissociation degree ($\alpha$). As $\alpha$ increased from 0.017% to 4.75%, the breaking strength increased from 35 to 117 MPa, fracture strain increased from 7.5% to 11.7%, and the toughness increased from 1.33 to 7.65 MJ m$^{-3}$; as $\alpha$ further increased to 58.4%, we observed a decrease in the breaking strength, fracture strain, and toughness. This indicates that the assembly of the polyelectrolyte polymer chains highly affected the mechanical properties for both fibers and films, which would be applicable for different types of polymer electrolytes, such as PAM and PDMAEMA.

We further investigated the alignment degree of the nanofibrils and polymer chains in the PAF$_\alpha$ by employing 2D SAXS and 2D WAXS, respectively (Fig. 3d, e, Supplementary Fig. 16). The orientation degree of the nanofibrils ($f_n$) and the orientation degree of the molecular chains ($f_m$) can be used to quantify the alignment degree of the polymer chains in the PAF$_\alpha$. Here, the orientation degree of the nanofibrils ($f_n$) was calculated from the azimuthal-integrated intensity distribution curves of the 2D SAXS patterns. The $f_m$ was calculated as

$f_m = (180° - \theta_h)/180°^{36}$, where $\theta_h$ is the angle of the half-width arc integrated from the diffraction intensity of the 2D WAXS pattern of the PAF$_\alpha$. As $\alpha$ increased from 0.028% to 4.75%, $f_n$ increased from 90.3% to 93.3%, and it then decreased to 91.1% as $\alpha$ increased further to 58.4% (Fig. 3d). As $\alpha$ increased from 0.028% to 4.75%, $f_m$ increased from 36.3% to 56.5%, and it then decreased to 39% as $\alpha$ increased further to 58.4% (Fig. 3e). Thus, modifying the dissociation of polymer chains by adding a base or an acid resulted in possibility of tailoring their flexibility. The moderate dissociation of PAF$_\alpha$ favors the alignment of the polymer chains, favoring the re-organization of the molecular chains to form nanofibrils.

We then calculated the free energy of polymer chain relaxation ($E_r$) of the PAF$_\alpha$, which was obtained by stretching the fiber to a pre-strain of 5% using a mechanical tester to measure the time it took for the stress to decrease to $1/e$ of the initial stress at different environmental temperatures[37] (Fig. 3f and Supplementary Fig. 17). The PAF$_\alpha$ showed slower stress relaxation for $\alpha$ of 0.17% and 4.75% than for $\alpha$ of 0.028% or $\alpha$ of 58.4%. For example, the stress relaxation time at 20 °C was 300 s, 166 s, and 36.9 s for $\alpha$ of 0.17%, 4.75%, and 58.4%, respectively. The $E_r$ increased from 32.3 to 50.5 kJ mol$^{-1}$ as $\alpha$ increased from 0.028% to 4.75%, and it then decreased to 40.4 kJ mol$^{-1}$ as $\alpha$ increased further to 58.4%. The dependence of polymer chain interactions on change of the dissociation degree $\alpha$ corresponds well with that of the size of the nanofibrils and alignment degree of polymer chains with different $\alpha$ values.

## Theoretical simulation

To theoretically understand the dependence of the dissociation of the polymer chains on the molecular chain flexibility, we carried out molecular dynamics (MD) calculations by employing the LAMMPS package[38,39] and the cluster analysis of different $\alpha$ systems via coarse-grained molecular simulations[40]. The MD simulation results indicate that the number of hydrogen bonds between the polyacrylic chains decreases with increasing dissociation degree (Supplementary Fig. 18a), which is consistent with the change of the calculated free energy of polymer chain relaxation as a function of dissociation degree. For the PAF$_{0\%}$, the formation of numerous hydrogen bonds between the carboxyl groups results in large multi-branched clusters, and the strong intermolecular interactions hinder the free migration of the chain segments and limit the mobility of the polymer chains. As $\alpha$ increased from 0% to 25%, the clusters transformed from a multi-branched shape to finely dissociated (Fig. 1b, c, Supplementary Data 1). The partial dissociation of the carboxyl groups decreased the density of the hydrogen bonds, and the cations dispersed between the polymer chains increased the intermolecular spacing of the polymer chains and improved the mobility of the polymer chain segments. This favored the ordered alignment of the polymer chains during stretching and dehydration. As $\alpha$ increased further to 50%, the volume of the clusters increased (Supplementary Fig. 19), and the originally dispersed clusters formed aggregates through the interaction of the metal ions with carboxylate, preventing the movement of the polymer chains. The cluster analysis results are in good agreement with the mean squared displacement (MSD) of the polymer centers of mass for different dissociation degrees (Supplementary Fig. 18b). The theoretical modeling results of molecular interactions agree well with the evolution of the nanofibrils structure as observed in SEM and AFM as well as the fiber mechanical strength with the change of the dissociation degree.

## Crack-resisting properties of the PAF artificial spider silk

The existence of axial nanofibrils in the PAF$_\alpha$ prompted us to investigate its capacity to work in the presence of notches (crack-resisting capacity) for different $\alpha$ values (Fig. 2e and Supplementary Figs. 20 and 21). A 15-μm-deep notch was fabricated in the radial direction of a PAF$_\alpha$ with a diameter of 100 μm; then the fiber was stretched using a mechanical tester at a stretch rate of 5 mm/min. A PAF$_\alpha$ without the notch was used as the control sample. The mechanical testing results show that the PAF$_{4.75\%}$ exhibited excellent notch-resisting capacity (Supplementary Fig. 20). The PAF$_{4.75\%}$ exhibited ultra-fine nanofibrils with average diameter of 17.7 nm. In this case, the crack propagation was effectively inhibited, and the nanofibrils were stretched out during mechanical stretching. For comparison, the PAF$_{0.017\%}$ and PAF$_{96.5\%}$ exhibited sharp cracking cross-section at fiber fracture. Here, the ratio of the breaking strength, breaking strain, and fracture toughness of the PAF$_\alpha$ with the notch to those of the PAF$_\alpha$ without the notch ($\sigma_n/\sigma_0$, $\varepsilon_n/\varepsilon_0$, and $\Gamma_n/\Gamma_0$, respectively) were employed to quantify the crack-resisting capacity. As $\alpha$ increased from 0.017% to 4.75%, $\sigma_n/\sigma_0$, $\varepsilon_n/\varepsilon_0$, and $\Gamma_n/\Gamma_0$ increased from 0.42, 0.52, and 0.25 to the maximum values of 0.78, 0.70, and 0.54, respectively, and then decreased with further increase in $\alpha$ (Fig. 2e).

A notch-containing PAF$_{4.75\%}$ showed an optimized breaking strength of 0.32 GPa and a fracture energy of 1.65 MJ m$^{-2}$. Such a combination of high strength and fracture energy is among the best ever reported for crack-resistant materials, such as poly(urethane-urea) elastomers (0.017 GPa and 0.21 MJ m$^{-2}$, respectively), poly(vinyl alcohol) nanocomposites (0.1 GPa and 0.13 MJ m$^{-2}$, respectively), poly(1-acrylamido-2-methylpropane sulfonic acid)/poly(N,N-dimethy-lacrylamide) ionogel (0.0003 GPa and 0.087 MJ m$^{-2}$, respectively), poly(acrylamide-co-acrylic acid) ionogel (0.012 GPa and 0.024 MJ m$^{-2}$, respectively), poly(sodium $p$-styrenesulphonate-co-(methacryloyla-mino)propyl-trimethylammonium chloride hydrogel (0.002 GPa and 0.004 MJ m$^{-2}$, respectively), and poly(2-(dimethylamino)ethylacrylate) methyl chloride quaternary salt/poly(methylacrylic acid) microfibers (0.010 GPa and 0.187 MJ m$^{-2}$, respectively) (Fig. 2f, Supplementary Table 8).

At appropriate dissociation degree (e.g. 4.75%), the polymer chains finely dissociated and are easily aligned with one another during the draw-spinning process. Then, during the water evaporation process, large amount of hydrogen bonding formed between the well-aligned polymer chains. The neighbor polymer chains got closely packed and self-assembled to form nanofibrils. Increasing the alignment degree of the polymer chains in the nanofibril would allow the fiber withstand heavier loading stress along the fiber axis. Bundles of the polymer chains formed a lot of highly aligned nanofibrils. Inside the nanofibirl strong interactions exist between the polymer chains, while the interaction between these nanofibrils are relatively weak. This can be confirmed by the fact that bundles of nanofibrils were pulled off from the longitudinal section of the fiber after fiber fracture. During mechanical stretching of a fiber, in the case that there is a defect on the fiber surface, the nanofibrils in the defect would break to form a crack. The crack would be stopped and would not propagate to the neighbor nanofibrils, because of the relatively weak interaction and less entanglement between the polymer chains in the neighbor nano-fibrils. While crack propagation and stress concentration in the defects of a common polymer material is considered as an important mechanical failure mechanism. Consequently, fracture of some nano-fibrils of a fiber would not result in immediate breakage of the fiber, which would still withstand the loading stress because of the integrity of the remaining nanofibrils. In addition, the sliding, unfolding, elongation of nanofibrils in the fiber rather than crack all together during mechanical stretching would also help increasing the mechanical strength and toughness. Therefore, we observed increased mechanical strength, strain, and toughness for the fiber with obviously fine nanofibrils that exhibited increased polymer chain alignment.

## Improving mechanical properties by drawing fiber through an extensional flow field

The hierarchical structure of spider silk was formed by the spidroin molecules experiencing a series of physical and chemical changes in ever-increasing mechanical stress fields, which contributed to the

super-strong mechanical performance of spider silk. Spiders draw spun spider silk directly from the spinning dope, which can directly produce a fiber in air with highly aligned molecular chains and nanofibrils. In order to further improve the mechanical properties of the PAF, we replicate the natural process by subjecting bulk polyacrylic acid gel to an extensional flow field to draw fibers by using a metal wire (Supplementary Fig. 22). A PAF artificial spider silk was directly drawn spun from a bulk gel. Then, the fiber was placed in a 60 °C oven to dry 3 h with both ends taped on a home-made frame. The relative humidity plays an important role in controlling the diameter of the PAF during draw-spinning. As RH increased from 10% to 50%, the diameter of $PAF_{0.17\%}$ increased from 5.6 μm to 35.2 μm, and the corresponding fiber sheath ratio decreased from 69.5% to the 50% (Supplementary Fig. 23). The microstripes and nanofibrils were also observed in the fiber surface and longitudinal sectional SEM images. The drawn-spun PAF shows a lower water content (1.9%), higher melting temperature (129.3 °C), and higher orientation degree of the nanofibrils (93.6%) than the PAF (5.4%, 125.0 °C, and 92.4%, respectively). This indicates that drawing spun a PAF artificial spider silk directly from a bulk gel enables the production of thinner and stronger fibers than those obtained from polymerization in a capillary tube.

We next optimized the mechanical properties of the draw-spun $PAF_\alpha$ with different $\alpha$ values for various stretch rates during the mechanical measurements and the twist density. The draw-spun $PAF_\alpha$ was obtained for $\alpha$ ranging from 0.17% to 2.43%, and the diameter was controlled as $5 \pm 1$ μm at a stretching rate of 20 mm/min under the relative humidity of 10%. As $\alpha$ increased from 0.17% to 1.1%, the breaking stress, fracture strain, and toughness increased from 0.63 GPa, 32%, and 136 MJ m⁻³ to the maximum values of 0.87 GPa, 42%, and 272 MJ m⁻³, respectively; they then decreased with a further increment in $\alpha$ (Fig. 2c, Supplementary Fig. 24). As the stretch rate increased from 100 to 500 mm min⁻¹, the breaking strength of the $PAF_{1.1\%}$ increased monotonically from 1.16 to 1.83 GPa, and its fracture strain decreased monotonically from 31% to 18.8%, with the maximum toughness of 262 MJ m⁻³ obtained at a stretch rate of 300 mm min⁻¹ (Supplementary Fig. 27). We then inserted twist into the PAF in the gel state and dried it to set the shape; this enables the realization of polymer chains with a spiral architecture by inducing a torsional stress in such a gel state[41]. As the twist was inserted in the fiber in the gel state, after twist insertion, both ends of the twisted fiber were tethered and the fiber was allowed to dry. Then, the inserted twist would be kept in the fiber because of formation of the hydrogen bonds between the polymer chains after water evaporation. We inserted twist in a 95-μm-diameter $PAF_{4.75\%}$ that was prepared by photopolymerization in a capillary tube. With increase of the twist density, the orientation degree of nanofibrils and the mechanical strength and toughness increased to a maximum value, and then followed by decrease with further increase in the inserted twist density (Supplementary Fig. 25). As the twist density increased from 0 to 1.0 turns mm⁻¹, the orientation degree increased from 86% to the maximum value of 88.7%, the breaking strength increased from 0.42 GPa to the maximum value of 0.50 GPa, and the toughness increased from 336 MJ m⁻³ to the maximum value of 498 MJ m⁻³; the orientation degree, breaking strength and toughness decreased with further increase in the twist density. We also insert twist in a drawn-spun $PAF_{1.1\%}$ and investigated its mechanical properties. With increase in the twist density, the alignment degree of nanofibrils and mechanical strength monotonically decreased, the breaking strain and toughness monotonically increased (Supplementary Fig. 26). This would result from the fact that the molecular chains are already well-aligned in the drawn-spun $PAF_{1.1\%}$, and twist insertion resulted in spiral configuration of the polymer chains. As the twist density increased from 0 to 16 turns mm⁻¹, the fracture strain of the $PAF_{1.1\%}$ increased monotonically from 18.8% to 35.8%, and the breaking stress decreased monotonically from 1.83 to 1.21 GPa, with the maximum toughness of 339 MJ m⁻³ obtained at the twist density of 12 turns mm⁻¹ (Fig. 2d).

By optimizing the flexibility of polymer chains, fiber preparation process, and characterization conditions, the $PAF_\alpha$ exhibited combination of breaking strength and toughness ranging from 1.83 GPa and 238 MJ m⁻³ for $\alpha$ of 1.1% (Fig. 2d) to 0.53 GPa and 700 MJ m⁻³ for $\alpha$ of 4.75% (Supplementary Fig. 8e, f). Such a combination of high mechanical strength and toughness is the best among the currently reported artificial fiber materials, such as chimeric protein fibers (0.60 GPa and 113 MJ m⁻³, respectively), amyloid protein fibers (0.86 GPa and 160 MJm⁻³, respectively), cellulose nanofibrils/spidroin protein fibers (0.47 GPa and 32 MJ m⁻³, respectively), hydroxyapatite/ polyvinyl alcohol/sodium alginate fibers (0.95 GPa and 296 J g⁻¹, respectively), poly(urethane-urea) fibers (0.07 GPa and 390 MJ m⁻³, respectively), and hydroxyapatite/polyurethane fibers (0.44 GPa and 640 MJ m⁻³, respectively) (Fig. 1e, Supplementary Table 7). The nanofibril size resulted excellent mechanical properties, which is among the best of the currently reported artificial fiber materials, such as supermolecular fiber (100 nm, 0.193 GPa, respectively), regenerated B. mori silk fiber (100 nm, 0.4 GPa, respectively), protein/genipin fiber (38.4 nm, 0.02 GPa, respectively), and regenerated spidroins fiber (100 nm, 0.3 GPa, respectively) (Supplementary Table 9).

## $PAF_\alpha$ artificial spider silk for artificial muscles

The spider silk exhibited moisture-driven supercontraction behavior, showing a decreased length in high-humidity environments[42]. During such a process, the highly aligned molecular chains that are frozen in the spider silk become highly flexible and transform into random coils. Inspired by such a phenomenon, moisture driven supercontraction was observed for the hydrogel fibers, which was considered for employing as artificial muscles[18,43]. Thermally driven polymer actuators were reported for various applications with the aid of metal nanowire heater, such as color-changing and anisotropic soft actuators[44], photomechanical nanowire actuators[45], and directional transparent shape morphing actuators[46]. By considering that the $PAF_\alpha$ fibers exhibited highly aligned molecular chains, and heat can disrupt the H-bonds to increase the molecular chain mobility, so that the molecular entropy would be possible to drive the highly aligned molecular chains to rearrange to lower energetic configuration. Until now the thermally driven supercontraction of hydrogel fibers was rarely reported.

We here report the thermally driven supercontraction behavior of the $PAF_\alpha$-based artificial spider silk (Fig. 4). A 95-μm-diameter $PAF_{4.75\%}$ prepared in a capillary tube with a pre-strain of 1200% was dried in air at an RH of 10% for 15 min to set the length. Heating this $PAF_{4.75\%}$ to 100 °C caused the fiber to contract by 88% (also called supercontraction degree) within 10 s to reach a plateau (Supplementary Fig. 29); this is ascribed to the increased molecular chain flexibility at an elevated temperature[47] and can be employed in the realization of irreversible artificial muscles for actuation. TGA showed good stability for the investigated thermal actuation temperature range (Supplementary Fig. 28). We also investigated the actuation stroke of the 30-μm-diameter drawn-spun $PAF_{1.1\%}$ at different temperatures between 40 °C to 100 °C without loading a mass (Supplementary Fig. 32). The drawn-spun $PAF_{1.1\%}$ contracted by 38.3% in 8.0 s at 40 °C and reach a plateau. The response time decreased from 8.0 s to 2.0 s as the actuation temperature increased from 40 °C to 100 °C, corresponding to a maximum actuation speed increasing from 12.2% s⁻¹ to 173.8% s⁻¹. We investigated the supercontraction performance of $PAF_\alpha$ with different $\alpha$ values, and also correlated with alignment degree of molecular chains and nanofibrils. As $\alpha$ increased from 0.017% to 4.75%, the actuation strain increased from 8.9% to the maximum value of 65%, and the work capacity increased from 0.2 J g⁻¹ to the maximum value of 1.5 J g⁻¹, by heating the $PAF_\alpha$ loaded with 2.7 MPa from 25 °C to 100 °C; at the same time, the orientation degree of molecular chains ($f_m$) increased from 36.3% to the maximum value of 56.5%, and the orientation degree of nanofibrils ($f_n$) increased from 90.3% to the maximum value of 93.3%; as $\alpha$ further increased to 21.9%, we observed decrease

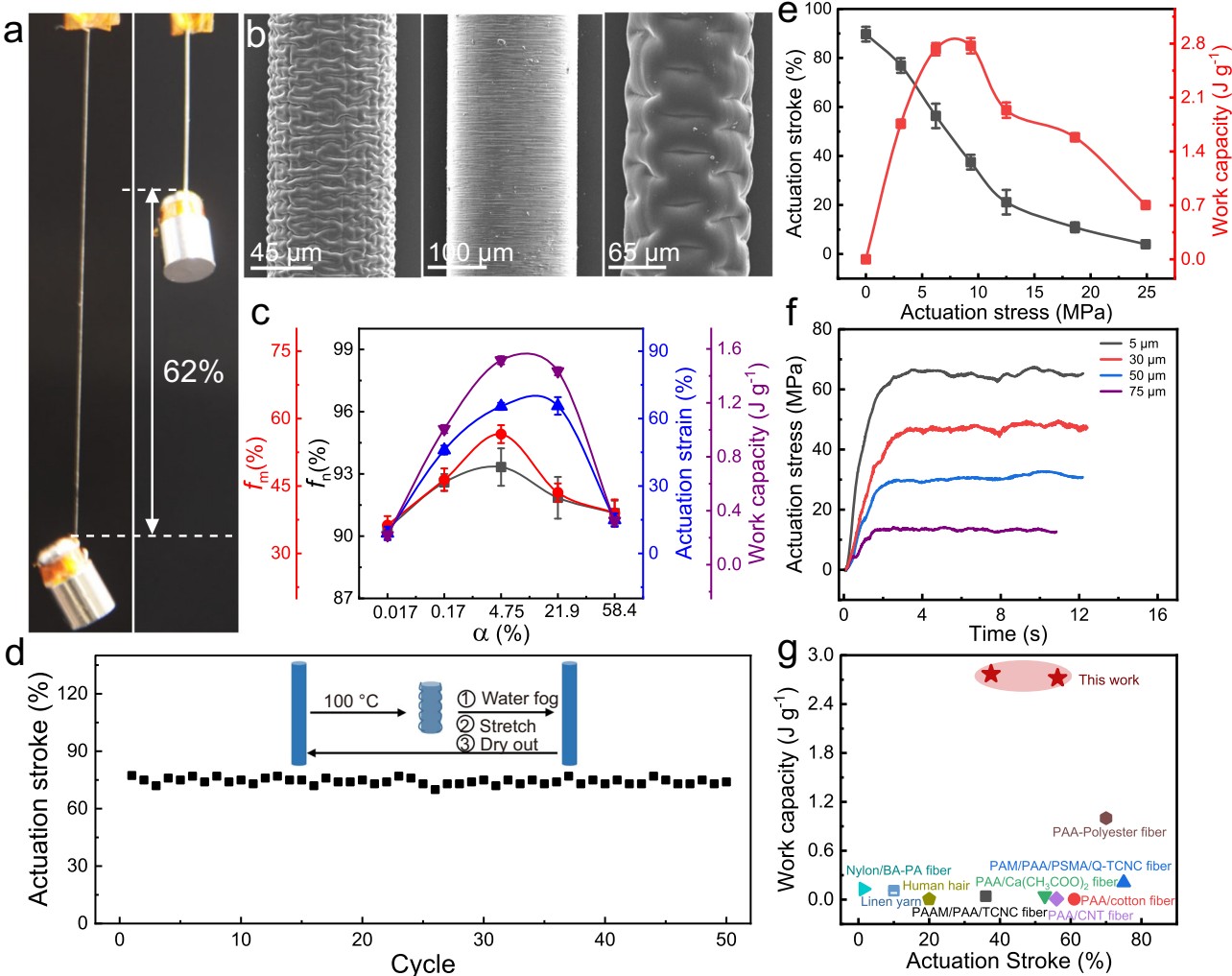

**Fig. 4 | Actuation properties of the PAF$_\alpha$-based artificial spider silk.**
**a** Photographs of a 95-µm-diameter PAF$_{4.75\%}$ before and after actuation (lifting a 2.0-g load). **b** SEM images of the PAF$_\alpha$ with different $\alpha$ values after actuation. **c** The dependence of the actuation strain, work capacity, the orientation degree of molecular chains ($f_m$), and the orientation degree of nanofibrils ($f_n$) of the PAF$_\alpha$ on $\alpha$ values. The PAF$_\alpha$ was loaded with 2.7 MPa by heating from 25 °C to 100 °C for actuation. **d** Actuation stroke of the PAF$_{4.75\%}$ under an isobaric load of 1.45 MPa as a function of the number of cycles. The inset shows the schematic of the process. **e** Actuation stroke and work capacity of a 5-µm drawn-spun PAF$_{1.1\%}$ as a function of

isobarically loaded stress. The actuation temperature was 100 °C. **f** Contractile actuation stress of the drawn-spun PAF$_{1.1\%}$ as a function of time by heating 25 °C to 120 °C by employing two-parallel heating plates measured on the mechanical tester. The drawn-spun PAF$_{1.1\%}$ was both-end tethered on the holder of the mechanical tester to obtain the actuation stress. **g** Comparison of the work capacity and actuation stroke of the drawn-spun PAF$_{1.1\%}$ prepared in this work with those of typical artificial hydrogel muscles reported in the literature. The error bars for **c** and **e** represent mean ± SD ($n = 3$ independent samples).

of the actuation strain, work capacity, $f_m$, and $f_n$ (Fig. 4c, Supplementary Fig. 30). Interestingly, a buckled surface was observed for the PAF$_\alpha$ after supercontraction, with the smallest buckle width (1.0 µm) obtained for $\alpha$ of 4.75% (Fig. 4b). The occurrence of surface buckling reflects the presence of a sheath–core structure; indeed, it is the mechanical mismatch between the rigid sheath and the elastic core during contraction that causes the surface buckling of the PAF$_\alpha$.

We inserted twist in the 95-µm-diameter PAF$_{4.75\%}$ that was prepared by photopolymerization in a capillary tube to different twist densities, and investigated the fiber's actuation performance at an actuation temperature of 100 °C under different loading stresses (Supplementary Fig. 31). As the twist density increased from 0 to 1.0 turns mm$^{-1}$ for the PAF$_{4.75\%}$ under the loading stress of 3.53 MPa, the actuation stroke increased from 40.9% to the maximum value of 58.5%, and the work capacity increased from 1.33 to the maximum value of 1.68 J g$^{-1}$; with further increase of the twist density, the actuation stroke and the work capacity decreased. This showed the same dependence of fiber mechanical strength on the twist density. There are two effects

of the polymer chain orientation by twist insertion of the PAF$_\alpha$ fiber in the gel state. The polymer chains exhibited a random coil morphology in the as prepared fiber. During twist insertion the polymer chains exhibited a spiral alignment under the torsional stress. The increased orientation of the polymer chains increased the mechanical stress and the actuation stress, while spiral architecture of the polymer chains resulted in a twist angle between the polymer chain orientation and the fiber axial direction, causing a decrement in the axial contribution of the mechanical stress and the actuation stress. Therefore, with increase of the inserted twist density, the work capacity of the PAF$_\alpha$ first increased to a maximum value and then decreased.

A 95-µm-diameter PAF$_\alpha$ loaded with 1.45 MPa mass at room temperature (25 °C) was heated to 100 °C to allow supercontraction to the maximum actuation stroke. Then, the PAF$_\alpha$ was cooled to room temperature, and the fiber would keep at this contracted length. The PAF$_\alpha$ was then exposed to ultrasonically generated water fog (95% relative humidity) and re-stretched to the initial length, followed by air-drying for 15 min to set the length. The above cycling process was repeated for

50 times. The schematic of the above process was shown in the inset of Fig. 4d.

We obtained optimized actuation stress by employing the drawn-spun $PAF_\alpha$ with different diameters, by directly measuring the actuation stress on the mechanical tester. This was realized by holding the both ends of the $PAF_\alpha$ on the clamp of the mechanical tester and heating the fiber to an elevated temperature via two-parallel heating plates. The drawn-spun $PAF_{1.1\%}$ with diameter of 75, 50, 30 and 5 μm were investigated. As the diameter of the $PAF_{1.1\%}$ decreased from 75 to 5 μm, the actuation stress increased from 13.6 to 65 MPa, which is in the same level of the actuation stress of the natural spider silk (~80 MPa)[48]. We further investigated the load-lifting capacity of the 5 μm-diameter drawn-spun $PAF_{1.1\%}$. The 5-μm-diameter drawn-spun $PAF_{1.1\%}$ can lift a 9.33 MPa load by an actuation stroke of 37.5%, corresponding to an actuation work capacity of 2.77 J g$^{-1}$; it also can lift a load of 18.6 MPa by an actuation stroke of 11%, corresponding to an actuation work capacity of 1.58 J g$^{-1}$ (Fig. 4e, f).

Because $PAF_\alpha$ has been dried in an oven at 60 °C before measuring the mechanical and actuation performance, it is not a hydrogel material. The combination of breaking strength (1.8 GPa) and actuation stress (65 MPa) of the $PAF_\alpha$ is among the best of the actuation materials including hydrogel materials and shape memory polymer materials, such as PAAM/PAA/cellulose nanocrystals fibers (0.065 GPa for breaking strength and 0.24 MPa for actuation stress), PAA/cotton fibers (0.15 GPa for breaking strength and 0.03 MPa for actuation stress), and PAA/carbon nanotube fibers (0.2 GPa for breaking strength and 0.01 MPa for actuation stress), COCe-PE bimorph fiber (0.0258 GPa for breaking strength and 5 MPa for actuation stress), block polymer (0.146 GPa for breaking strength and 5.5 MPa for actuation stress), LCE-Graphene fiber (0.09 GPa for breaking strength and 1.23 MPa for actuation stress) (Fig. 4g and Fig. 1f, Supplementary Tables S10 and S11).

The heating-induced supercontraction of $PAF_\alpha$ should be ascribed to the morphology change of the highly oriented molecular chains (exhibiting low entropy state) to the low oriented molecular chains (exhibiting high entropy state). This process is similar to the shape memory behavior, and the proposed mechanism is as follows. The as-prepared $PAF_\alpha$ after polymerization contains a large amount of water molecules, and the hydrogen bonding between the polymer chains was disrupted by the water molecules. Therefore, the fiber exhibited high elasticity, and the molecule chains exhibited low orientation and high entropy. Then, the as-prepared $PAF_\alpha$ was pre-stretched to an elongated length and air dried. The molecular chains after pre-stretch exhibited high orientation and low entropy. After water evaporation, the hydrogen bonding between the polymer chains was re-constructed, and the $PAF_\alpha$ kept at this elongated length. By heating, the hydrogen bonding between the polymer chains was disrupted at an elevated temperature, and the polymer chains spontaneously changed to the morphology with low orientation degree (exhibiting high entropy). Supplementary Fig. 33 shows the 2D WAXS patterns before and after thermally driven supercontraction, which shows decreased anisotropy and alignment degree of molecular chains after thermally driven supercontraction.

Different from the common shape-memory polymer materials, after thermally driven supercontraction, the hydrogel-based $PAF_\alpha$ can be easily re-strained to the elongated length after wetting, followed by air-dry to set the shape for the next round of supercontraction by heating. Such a process is a synergistic combination of entropy-driven morphology change of polymer chains and the breaking and re-formation of hydrogen bonding between the polymer chains of the $PAF_\alpha$. Negligible decay of the actuation performance of the $PAF_\alpha$ was observed for 50 cycles of thermally driven supercontraction (Fig. 4d). Moreover, because the $PAF_\alpha$ exhibited spider-silk-like hierarchical structure with tunable nanofibrils of the polymer chains, we further optimized the supercontraction capacity of the $PAF_\alpha$, which exhibited excellent actuation performance with maximum actuation stress up to 65 MPa (Fig. 4f).

To achieve two-way reversible actuation, twist was inserted in the $PAF_\alpha$ to form self-coil or mandrel-coil architectures (Supplementary Figs. 34 and 35). Briefly, taking the $PAF_{4.75\%}$ as an example, the $PAF_{4.75\%}$ was polymerized in a capillary tube, prestrained to 1200%, and air-dried for 15 min at RH 10% to obtain a 95-μm-diameter fiber. To prepare a self-coiled artificial muscle, we insert twist (4.5 turns mm$^{-1}$) into a 95-μm-diameter $PAF_\alpha$ until fiber coiling under a constant load of 3.3 MPa, followed by cross-linking by employing $Zr^{4+}$ to set the coiled shape to obtain a self-supporting artificial muscle with spring index of 1.17. We then measured the actuation stress of the self-coiled $PAF_{4.75\%}$ by tethering the both-ends of the sample on the mechanical tester. By heating the self-coiled $PAF_{4.75\%}$ artificial muscle from 25 °C to 120 °C, an actuation stress of 9.0 MPa was obtained, which was repeated for several heating-cooling cycles (Supplementary Fig. 34).

In addition, we also prepared mandrel-coiled artificial muscles by wrapping the twisted $PAF_{4.75\%}$ (2.0 turns mm$^{-1}$) around a mandrel, followed by cross-linking via $Zr^{4+}$ to set the coiled shape (Supplementary Fig. 35). The actuation performances were investigated for the homochiral $PAF_{4.75\%}$ artificial muscles with different twist densities and spring index. The mandrel-coiled $PAF_{4.75\%}$ artificial muscles exhibited high reversibility during repeated heating and cooling cycles. We measured the dependence of actuation stroke of the mandrel-coiled self-supporting $PAF_{4.75\%}$ artificial muscle with different twist densities and spring indexes without a load. The actuation stroke increased from 25% to 65% as the spring index increased from 10 to 50 for the mandrel-coiled $PAF_{4.75\%}$ artificial muscle with twist density of 2.0 turns mm$^{-1}$. The actuation stroke increased from 20% to 65% as the twist density increased from 1.0 to 3.0 turns mm$^{-1}$ for the mandrel-coiled $PAF_{4.75\%}$ artificial muscle with spring index of 30.

## Discussion

In summary, an artificial spider silk was prepared by tuning the dissociation degree of the $PAF_\alpha$ to modulate its molecular chain flexibility and self-assembly, establishing superfine nanodomain size (5.2 nm), achieving extraordinary combination of breaking strength of 1.83 GPa and toughness of 238 MJ m$^{-3}$. In addition, the $PAF_\alpha$ exhibited an exceptional thermally driven supercontraction behavior, and the establishment of the nanofibrils highly increased the actuation performance, reaching a work capacity of 2.77 J g$^{-1}$ and an actuation stress of 65 MPa. The design strategy of the $PAF_\alpha$-based artificial spider silk proposed in this work could be beneficial to numerous different applications, including energy absorption and damping, high-performance composites, space exploration, and the landing on the Moon or Mars. The actuation capacity of this artificial spider silk could enable the realization of multifunctional artificial muscles, smart fibers, soft robotics, and other bionic applications. The strategy of tailoring the polymer chain flexibility to modify the nanofibrils would inspire design possibilities in 3D printing, biomimetic smart materials, and hierarchical structural materials.

## Methods

### Preparation of the $PAF_\alpha$-based artificial spider silk

The $PAF_\alpha$ was prepared through the free-radical polymerization of a mixture solution of acrylic acid (40 wt%), sodium hydroxide (or hydrochloric acid), and photo initiator ammonium persulfate (with a mass ratio of 0.2:100 to the acrylic acid) in a 0.5-mm-diameter capillary tube under 365-nm UV-light irradiation for 2 h. Then, the capillary tube was broken in the middle to extract the hydrogel fiber, which was stretched to impose the desired strain and dried for 3 h at 60 °C with both ends tethered on a homemade frame to obtain the $PAF_\alpha$. The draw-spun PAF was prepared by dipping a 0.1-mm-diameter metal wire into the above bulk gel, which was then extracted at a speed of 4 cm s$^{-1}$.

## Preparation of the p(AM-co-DMAEMA) fiber

AM and DMAEMA with a molar ratio of 12:1 were employed as the monomers, silica nanoparticles functionalized with vinyl groups were employed as the crosslinker (SNV, with a mass ratio of 0.6:100 to the monomer; the total mass content of the monomer was 23%), and ammonium persulfate (APS) was employed as the initiator (with mass ratio of 0.5:100 to the monomer). The polymerization process was carried out for 5 min at 40 °C oven. The SNV was prepared through the hydrolysis of vinyltriethoxysilane (11.2 wt%) in distilled water for 12 h under stirring at room temperature.

## Preparation of the polyacrylic acid films with different α values

The bulk polyacrylic acid gel ($10 \times 10 \times 0.2$ mm$^3$) was biaxially stretched to form a film with a thickness of 12 μm and then air dried for 12 h at RH = 35% with the film tethered on a home-made 20-mm-diameter circular frame. The film was then employed for AFM characterization.

## Statistical analysis

Each experiment was repeated at least three times independently with similar results. All data are expressed as the mean ± standard deviation (SD). NanoScope Analysis 1.8 software was used for AFM images analysis. ImageJ software was used for SEM images analysis. Fit 2D software was used for 2D SAXS analysis. The general area detector diffraction system (GADDS) software was used for 2D WAXS analysis.

## Data availability

The data that support the findings of this study are available within the article and Supplementary Information files, or available from the corresponding authors on request. Source data are provided with this paper.

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

## Acknowledgements

This work was supported by the National Key Research and Development Program of China (Grants Nos. 2022YFB3807103, 2022YFA1203304, and 2019YFE0119600), the National Natural Science Foundation of China (grants 52350120, 52090034, 52225306, 51973093, 51773094, and 22371300, Z.F.L. and X.Z.), Frontiers Science Center for Table Organic Matter, Nankai University (grant number 63181206. Z.F.L.), the Fundamental Research Funds for the Central Universities (grant 63171219. Z.F.L.), Lingyu Grant (2021-JCJQ-JJ-1064, Z.L.F.), and Beijing-Tianjin-Hebei Basic Research Cooperation Project (No. J230023, E.Z.L.). This work was also supported by the User Experiment Assist System of Shanghai Synchrotron Radiation Facility (SSRF) and Beijing Synchronization Radiation Facility (BSRF).

## Author contributions

Z. Liu, X. Zhou, M. Zhu, and W. He conceived the project. W. He, M. Wang, S. Liu, A. Khan, C. Li, D. Feng, Z. Su, L. Bao, G. Wang, E. Liu, Y. Zhu, J. Bai carried out the experiments, characterization, and data analyses. G. Mei contributed to theoretical simulation and calculation. All authors wrote the paper. All authors provided comments and agreed with the final form of the manuscript.

## Competing interests

The authors declare no competing interests.
