## [Peer Review File · Nature Communications]

Establishing Superfine Nanofibrils for Robust Polyelectrolyte Artificial Spider Silk and Powerful Artificial MusclesREVIEWER COMMENTS

Reviewer #1 (Remarks to the Author):

This paper presents the development of superfine nanofibrils in polyelectrolyte artificial spider silk by optimizing the flexibility of polymer chains and by introducing ions to control the dissociation of polymer chains and evaporation induced self-assembly under external stress.

This work could provide inspiration for the design of high-performance fiber materials and can be recommended for publication with the following comments,

1) The polypeptide chains of the spidroins exhibited high flexibility in the spinning dope of the spider, which self-assembles into fine nanofibrils during spinning into the spider silk exhibiting hierarchical structure. The authors investigated the establishment of superfine nanofibrils of polyelectrolyte artificial spider silk by optimizing the polymer chain flexibility (Fig. 1). The artificial spider silk was prepared from hydrophilic polymer electrolytes, such as poly(acrylic acid) (PAA), polyacrylamide (PAM), and poly(2-(Dimethylamino)ethyl methacrylate) (PDMAEMA). The authors may compare the artificial spider silk and artificial spider skin they synthesized.

2) To obtain a dry PAF, the wet PAF was stretched to thirteen times its initial length (1200% strain) and then dried in an oven at 60 °C (Supplementary Fig. 1). The PAF retained its elongated length for a drying time longer than 5 min. Supplementary Fig. 2 showed the obvious sheath-core structure with increasing drying time. Accompanying with the hydrogen bonding network between polymer chains, this provides a good platform for establishing hierarchical structures for polymer artificial spider silk. What is the function of sheath-core structure?

3) The authors mentioned that the aligned nanofibrils is key for improving the fiber mechanical properties, which can effectively increase the fiber strength and inhibit crack propagation during deformation. Therefore, the authors attempted to modulate the size and alignment of the nanofibrils in the hierarchically structured artificial spider silk by tuning the molecular chain flexibility to enhance the mechanical properties of the fibers. How did the aligned nanofibrils increase the fiber strength? And how was the degree of alignment measured?

4) The draw-spun PAF_α was obtained for α ranging from 0.17% to 2.43%, and the diameter was controlled as $5 \pm 1 \mu\text{m}$ at a stretching rate of 20 mm/min under the relative humidity of 10%. As α increased from 0.17% to 1.1%, the breaking stress, fracture strain, and toughness increased from 0.63 GPa, 32%, and 136 MJ/m³ to the maximum values of 0.87 GPa, 42%, and 272 MJ/m³. Does the relative humidity of 10% play an important role in this draw spun process?

5) A unique thermally driven supercontraction behavior was observed for the PAF-based artificial spider silk. Interestingly, the establishment of superfine nanofibrils provided the obtained PAF-based artificial spider silk highly increased actuation properties. The PAF exhibited an actuation work capacity of 1.5 J/g, an actuation stroke of 65%, and an actuation stress of 2.7 MPa at optimized dissociation degrees. Similar thermally driven actuators have been developed for various applications with the aid of metal nanowire heater (Nanoscale, 7, 6457 (2015); Adv. Funct. Mater., 28, 1801847 (2018); Soft Robotics, 6, 760 (2019)). They need to be briefly discussed in the manuscript.

6) How was the speed for the actuation?

7) To theoretically understand the dependence of the dissociation of the polymer chains on the

molecular chain flexibility, molecular dynamics (MD) calculations were carried by employing the LAMMPS package and the cluster analysis of different α systems via coarse-grained molecular simulations. More detailed information on the MD calculation need to be provided in the supporting information section.

8) Some of the letters in some figure (for example, Figure 4f) are too small to read. They should be enlarged for enhanced visibility.

9) The usage of general language including typos and grammatical errors need to be checked again.

10) Some of the digital figure are missing scale bars. They should be added to the pictures.

Reviewer #2 (Remarks to the Author):

This paper reports a strategy of controlling nanofibril structure to create a powerful artificial muscle combining excellent mechanical and actuation performance. In the field of artificial muscles, it has always been a long-standing challenge to simultaneously realize high strength and work capacity, for example, increasing crosslinking density can improve mechanical strength but usually reduce actuation strain and work capacity. In this work, the reported method has successfully addressed such long-lasting challenge, representing a breakthrough in artificial muscle research. On the other hand, even though nanofibril structure has been reported in other polymer materials showing high strength and toughness, to the best of the reviewer's knowledge, these systems such as polyvinyl alcohol microfibers usually don't exhibit excellent actuation performance. Furthermore, in terms of actuation performance, it is very rare for soft actuators to simultaneously show such high work capacity, actuation stroke, actuation stress and work capacity. Based on these reasons, I would like to recommend the publication of this manuscript, provided that the following comments can be addressed.

1. The key to this material design is the introduction of nanofibril structure, and the authors have found the significant mechanical properties enhancement as a result. However, the reviewer is still confused about the toughening mechanism even after finishing reading the manuscript. The authors should provide more explanations.

2. Is inserted twist structure stable? Because in most cases, the elasticity will result in complete untwist after removing the torsional tethering.

3. How does the twist insertion affect the orientation degree? And how does it affect the mechanical properties?

4. I suggest the authors to compare the actuation performance of twisted fiber to that of aligned fiber, which should be important for designing actuators with improved performance.

5. The fiber is actuated through shape memory mechanism, therefore, I think the authors should compare the mechanical properties and actuation performance to other thermoresponsive shape

memory polymers, such as Nat. Nanotechnol. 2022, 17, 1198; and ACS Central Science 7, no. 10 (2021): 1657-1667. As the excellent mechanical and actuation performance have also been reported in both examples.

6. Is it possible to realize two-way reversible actuation? It will be more desirable for the real-world applications.

7. How about the actuation performance at lower actuation temperature?

Reviewer #3 (Remarks to the Author):

This work prepared a polyelectrolyte artificial spider silk with thermal-induced supercontraction ability. It is indicated that the high-performance manmade fiber can be used as artificial muscle, and the rope made from silk can pull a car. I am sure that many existing ropes can withstand the car pulling to some extent. However, this work is not presented logically and some fundamental things are wrong and the data for the presented figures are not complete. Thus, I don't think it is suitable for publication in Nature Communications. Here are some concerns and questions:

1. What's the definition of artificial spider silk? This fiber does not share a similar structure to the natural spider silk.

2. It is extremely misleading in Figure 1, where a 4-ply-rope with 3000 fibers and twists is compared with a single spider silk fiber in literature. The images and charts for comparison there are not fair and not logical. Their so called high-performance manmade fiber can be used as artificial muscle, and the rope made from silk can pull a car. I am sure that many existing ropes can withstand the car pulling to some extent. Thus, this material is not that magical as they claimed.

3. How the authors test the mechanical properties (for example, on a single silk, ply silk, two-ply, four-ply, or rope)? This should be indicated in the manuscript otherwise the data are not complete. Besides, there are also many basic charts are not presented such as stress-strain curves of tensile testing, thus collected data can not be reasoned.

4. How to calculate work capacity? How to make a notch on the fiber? How to get the actuation stress? The author mentioned inserting twist in the fiber. What's the twisting parameter? The Method section should be elaborated in detail.

5. The supercontraction presented in the article is a one-time response. How to get a continuous cycle of the tensile stroke in Figure 4e. The actual recycling data there may be very rough, but the presentation seems to be very effective. This is also a misleading factor.

6. The author explained that the high breaking strength and toughness of the fibers are achieved by introducing ions to control the dissociation of polymer chains and evaporation induced self-assembly under external stress. This part is investigated in detail. However, I want to question how the dissociation degree (α) influences the supercontraction behavior. What is the mechanism for the supercontraction. Contraction upon heating is common in many synthetic polymers and how to distinguish the supercontraction in this fiber from the others? It is mentioned on Page 12 that "Until now the thermally

driven supercontraction of hydrogel fibers was rarely reported.” And how about other synthetic fibers?

7. The actuation stress is 2.7 MPa only, which is extremely low, how could the authors say this is powerful muscle? The actuation stress of natural spider silk can reach 80 MPa. Data in Figure 4 seems to conflict with the mechanical properties mentioned in the other part. Figure 4a shows a silk lifting a load. Is this a single PAF, or PAF with twist, or plied PAF? For example, Figure 4a shows a silk can lift a 2g load with a stroke of 62%. However, when the load reaches 8 MPa, there is no tensile stroke (Figure 4c), which seems the actuation force is very low. It does not match their claim and data for high strength of the fiber.

8. The authors present so much data in the manuscript, and these data are short of backing evidences for understanding.

Responses to the Reviewer #1:

This paper presents the development of superfine nanofibrils in polyelectrolyte artificial spider silk by optimizing the flexibility of polymer chains and by introducing ions to control the dissociation of polymer chains and evaporation induced self-assembly under external stress. This work could provide inspiration for the design of high-performance fiber materials and can be recommended for publication with the following comments.

Response: We thank the reviewer for these insightful comments, and we have revised the manuscript according to these comments.

1) The polypeptide chains of the spidroins exhibited high flexibility in the spinning dope of the spider, which self-assembles into fine nanofibrils during spinning into the spider silk exhibiting hierarchical structure. The authors investigated the establishment of superfine nanofibrils of polyelectrolyte artificial spider silk by optimizing the polymer chain flexibility (Fig. 1). The artificial spider silk was prepared from hydrophilic polymer electrolytes, such as poly(acrylic acid) (PAA), polyacrylamide (PAM), and poly(2-(Dimethylamino)ethyl methacrylate) (PDMAEMA). The authors may compare the artificial spider silk and artificial spider skin they synthesized.

Response: We thank the reviewer for this valuable suggestion. We added experiments and discussions about the mechanical properties and nanofibril morphology of the hydrophilic polymer polyelectrolyte fibers and films. The revision is as follows.

In order to observe smaller nanodomains, we prepared flat polyacrylic acid films with different α values and characterized their surface using AFM (Figure 3c and Figure S14–15). The polyacrylic acid film was obtained by biaxially stretching the bulk gel to form a film followed by air drying for 12 h while tethering on a circular frame. The nanodomains were observed for the dried film under AFM. The PAF film showed a uniform morphology with negligible nanodomain for α of 0.017%, and distinct nanodomains were observed as α increased to 0.044%. The nanodomains became more delicate with further increase in α , and the average size decreased to a minimum average value of 5.2 nm at α of 4.75%, approaching to that of the nature spider silk (~ 4 nm)²⁰. With further increase of α , the nanodomain size dramatically increased, and finally

disappeared at α of 96.5%. These results further confirmed that modulating the dissociation degree of polymer chains for high flexibility can achieve delicate nanodomain structure, thus improving the mechanical properties of the artificial silk and film.

We measured the mechanical properties of the PAA films by adjusting the dissociation degree (α). As α increased from 0.017% to 4.75%, the breaking strength increased from 35 to 117 MPa, fracture strain increased from 7.5% to 11.7%, and the toughness increased from 1.33 to 7.65 MJ m⁻³; as α further increased to 58.4%, we observed a decrease in the breaking strength, fracture strain, and toughness. This indicates that the assembly of the polyelectrolyte polymer chains highly affected the mechanical properties for both fibers and films, which would be applicable for different types of polymer electrolytes, such as PAM and PDMAEMA. (Page 8, Line 215–237).

Figure S15. (a) AFM phase images of the surface of polyacrylic acid films ($20 \text{ mm} \times 20 \text{ mm} \times 12 \text{ }\mu\text{m}$) with different α values that were prepared by bi-axial stretching of a polyacrylic gel ($10 \text{ mm} \times 10 \text{ mm} \times 0.2 \text{ mm}$); (b) The stress–strain curves and (c) breaking stress, fracture strain, and toughness of polyacrylic acid films with different α . The strain rate is 50 mm min^{-1} .

2) To obtain a dry PAF, the wet PAF was stretched to thirteen times its initial length

(1200% strain) and then dried in an oven at 60 °C (Supplementary Fig. 1). The PAF retained its elongated length for a drying time longer than 5 min. Supplementary Fig. 2 showed the obvious sheath-core structure with increasing drying time. Accompanying with the hydrogen bonding network between polymer chains, this provides a good platform for establishing hierarchical structures for polymer artificial spider silk. What is the function of sheath-core structure?

Response: We thank the reviewer for this valuable comment. The PAF exhibited a sheath-core structure under the reflection mode of the Metallographic microscope (Figure S2a). The fiber sheath contains less water and shows less transparent, and therefore exhibited a high modulus. The fiber core is highly transparent and contains large amount of water, and therefore exhibited a low modulus and elasticity. Such a sheath-core structure provides a rigid-yet-flexible architecture. Accompanying with the hydrogen bonding network between polymer chains, this provides a good platform for establishing hierarchical structures for polymer artificial spider silk. With increase of the drying time from 5 to 60 min, we observed an increment of the fiber sheath and a decrement of the fiber core, with fiber sheath-ratio increased from 31% to 77%. As the drying time increased from 0.5 h to 3 h, the water content decreased monotonically from 11.7% to 5.4% and reached a plateau. The breaking strength of the PAF increased monotonically from 0.25 GPa to reach a plateau value of 0.37 GPa, and the PAF exhibited the maximum fracture strain (86%) and toughness (200 MJ m^{-3}) at a water content of 8.9% for a drying time of 2 h; both the fracture strain and toughness decreased with a further decrease in the water content (Figure S2b). We added the above discussion in the context. (Page 4, Line 108–124).

Figure S2. (a) The sheath–core structure of PAF for different drying time. As the drying time increased from 5 min to 1 h, the fiber sheath ratio increased from 31% to 77%, and the fiber almost completely dried at 2 h. (b) Breaking stress, breaking strain, toughness, and water content of PAF_{0.17%} dried at 60°C for different time.

3) *The authors mentioned that the aligned nanofibrils is key for improving the fiber mechanical properties, which can effectively increase the fiber strength and inhibit crack propagation during deformation. Therefore, the authors attempted to modulate the size and alignment of the nanofibrils in the hierarchically structured artificial spider silk by tuning the molecular chain flexibility to enhance the mechanical properties of the fibers. How did the aligned nanofibrils increase the fiber strength? And how was the degree of alignment measured?*

Response: We thank the reviewer for the valuable comments.

The nanofibrils, especially the nanodomain size of 2–6 nm achieved by self-assembly of the polypeptide chains in natural spider silk is considered as an important origin for its excellent mechanical strength and toughness^{20, 21}. Similarly, the mechanical properties of a fiber depend on the self-assembled structure^{17, 22, 23} and degree of alignment of the molecular chains²⁴⁻²⁶. The ordered arrangement of molecular chains can increase the fiber mechanical strength, and the self-assembled nanofibrils can inhibit crack propagation during deformation, thereby increasing energy dissipation²⁷⁻²⁹. (Page 2, Line 57–64)

The PAF_{4.75%} exhibited ultra-fine nanofibrils with average diameter of 17.7 nm. In this case, the crack propagation was effectively inhibited, and the nanofibrils were

stretched out during mechanical stretching. For comparison, the PAF_{0.017%} and PAF_{96.5%} exhibited sharp cracking cross-section at fiber fracture. We added the above discussion and results in the context of the manuscript. (Page 10, Line 297–301)

At appropriate dissociation degree (e.g. 4.75%), the polymer chains finely dissociated and are easily aligned with one another during the draw-spinning process. Then, during the water evaporation process, large amount of hydrogen bonding formed between the well-aligned polymer chains. The neighbor polymer chains got closely-packed and self-assembled to form nanofibrils. Increasing the alignment degree of the polymer chains in the nanofibril would allow the fiber withstand heavier loading stress along the fiber axis. Bundles of the polymer chains formed a lot of highly aligned nanofibrils. Inside the nanofibril strong interactions exist between the polymer chains, while the interaction between these nanofibrils are relatively weak. This can be confirmed by the fact that bundles of nanofibrils were pulled off from the longitudinal section of the fiber after fiber fracture. During mechanical stretching of a fiber, in the case that there is a defect on the fiber surface, the nanofibrils in the defect would break to form a crack. The crack would be stopped and would not propagate to the neighbor nanofibrils, because of the relatively weak interaction and less entanglement between the polymer chains in the neighbor nanofibrils. While crack propagation and stress concentration in the defects of a common polymer material is considered as an important mechanical failure mechanism. Consequently, fracture of some nanofibrils of a fiber would not result in immediate breakage of the fiber, which would still withstand the loading stress because of the integrity of the remaining nanofibrils. In addition, the sliding, unfolding, elongation of nanofibrils in the fiber rather than crack all together during mechanical stretching would also help increasing the mechanical strength and toughness. Therefore, we observed increased mechanical strength, strain, and toughness for the fiber with obviously fine nanofibrils that exhibited increased polymer chain alignment. (Page 11–12, Line 319–342)

Figure S20. (a) SEM images of the notched PAF_α after breakage and (b) the fracture process of the notched PAF_α during stretching.

Figure S21. Stress–strain curves of the original and notched PAF_α for different α values. The notch is $15\ \mu\text{m}$ and the fiber diameter is $100\ \mu\text{m}$. The stretch rate is $5\ \text{mm}\ \text{min}^{-1}$.

The orientation degree of the nanofibrils (f_n) was calculated from the azimuthal-integrated intensity distribution curves of the 2D SAXS patterns using the following equations:

$$f_n = \frac{3(\cos^2 \Phi) - 1}{2},$$

and

$$\langle \cos^2 \Phi \rangle = \frac{\int_0^{\pi/2} I(\Phi) \cos^2 \Phi \sin \Phi \, d\Phi}{\int_0^{\pi/2} I(\Phi) \sin \Phi \, d\Phi},$$

where Φ is the azimuthal angle, and $I(\Phi)$ is the one-dimensional intensity distribution along Φ . $\langle \cos^2 \Phi \rangle$ is calculated by integrating the intensity of the specific 2θ diffraction peak along Φ . We added this in the Supporting Information. (Page 3, Line 67–75)

Figure S16. (a) Azimuthal-integrated intensity distribution curve from 2D SAXS of PAF_α with different α values.

4) The draw-spun PAF_α was obtained for α ranging from 0.17% to 2.43%, and the diameter was controlled as $5 \pm 1 \mu\text{m}$ at a stretching rate of 20 mm/min under the relative humidity of 10%. As α increased from 0.17% to 1.1%, the breaking stress, fracture strain, and toughness increased from 0.63 GPa, 32%, and 136 MJ/m³ to the maximum values of 0.87 GPa, 42%, and 272 MJ/m³. Does the relative humidity of 10% play an important role in this draw spun process?

Response: We thank the reviewer for the valuable comment. The relative humidity plays an important role in controlling the diameter of the PAF during draw-spinning. As RH increased from 10% to 50%, the diameter of $PAF_{0.17\%}$ increased from 5.6 μm to 35.2 μm , and the corresponding fiber sheath ratio decreased from 69.5% to the 50%.

We have added this discussion in the context. (Page 12, Line 354–358)

Figure S23. (a) Metallographic microscopy images of the drawn-spun PAF_{0.17%} fiber at the different RH. Scale bar: 15 μm. (b) The fiber diameter and sheath ratio of the drawn-spun PAF_{0.17%} fiber at the different RH.

5) A unique thermally driven supercontraction behavior was observed for the PAF-based artificial spider silk. Interestingly, the establishment of superfine nanofibrils provided the obtained PAF-based artificial spider silk highly increased actuation properties. The PAF exhibited an actuation work capacity of 1.5 J/g, an actuation stroke of 65%, and an actuation stress of 2.7 MPa at optimized dissociation degrees. Similar thermally driven actuators have been developed for various applications with the aid of metal nanowire heater (*Nanoscale*, 7, 6457 (2015); *Adv. Funct. Mater.*, 28, 1801847 (2018); *Soft Robotics*, 6, 760 (2019)). They need to be briefly discussed in the manuscript.

Response: We thank the reviewer for the valuable suggestion. Thermally driven polymer actuators were reported for various applications with the aid of metal nanowire heater, such as color-changing and anisotropic soft actuators⁴⁴, photomechanical nanowire actuators⁴⁵, and directional transparent shape morphing actuators⁴⁶. We added this discussion in the context of the revised manuscript. (Page 15, Line 424–427)

6) How was the speed for the actuation?

Response: We thank the reviewer for this valuable question.

We investigated the actuation stroke of the 30- μm -diameter drawn-spun PAF_{1.1%} at different temperatures between 40°C to 100°C without loading a mass. The drawn-spun PAF_{1.1%} contracted by 38.3% in 8.0 s at 40°C and reach a plateau. The response time decreased from 8.0 s to 2.0 s as the actuation temperature increased from 40°C to 100°C, corresponding to a maximum actuation speed increasing from 12.2% s⁻¹ to 173.8% s⁻¹. We have added the experimental results and discussions in the context. (Page 15, Line 440–445)

Figure S32. (a) The actuation strain of drawn-spun PAF_{1.1%} as a function of time at different actuation temperatures. (b) The maximum actuation speed of the drawn-spun PAF_{1.1%} as a function of actuation temperature. The diameter of PAF_{1.1%} is 30 μm .

7) To theoretically understand the dependence of the dissociation of the polymer chains on the molecular chain flexibility, molecular dynamics (MD) calculations were carried by employing the LAMMPS package and the cluster analysis of different α systems via coarse-grained molecular simulations. More detailed information on the MD calculation need to be provided in the supporting information section.

Response: We thank the reviewer for this good suggestion. We added the detailed description about MD calculation in the Supporting Information, as follows. (Supporting Information, Page 7–8, Line182–212)

MD Simulations

The MD simulations were carried out using the LAMMPS software package.⁴⁻⁵

A Nosé–Hoover thermostat and barostat were employed to execute simulations in the NPT (constant pressure and constant temperature) ensemble, which was used to obtain a reasonable density as well as the box size to simulate the system in the canonical ensemble (with the same thermostat and no barostat). Specifically, a temperature damping parameter of 0.5τ (reduced time unit) was employed to maintain the temperature at $1.25T^*$, where the reduced temperature $T^* = 1 kT/\varepsilon$, k is the Boltzmann constant, T is the true temperature, and ε is an energy parameter. A pressure damping parameter of 5τ was employed to maintain the pressure at zero. Periodic boundary conditions and the velocity Verlet algorithm with a time step of 0.005τ were used in all the simulations.

The simulation details were as follows. Firstly, the soft potential was used to run the simulation for 1000τ to eliminate the particle overlap of the initial model under the NVT ensemble. Then, the soft potential was replaced by a mixed LJ and Coulombic potential, and each system was equilibrated for $5 \times 10^4\tau$ under the NPT ensemble to allow the density to reach a stable state. Subsequently, the barostat was removed, and the system was equilibrated for $2 \times 10^5\tau$ under the NVT ensemble. After a long relaxation time, the MSD of the molecular chains became greater than $3R_g^2$, indicating that the structure of the aggregation state had become relatively stable. Finally, the MSD (Figure S18b) calculation and cluster analysis were performed on the basis of the stable structure.

The cluster was defined as a group of closely aggregated particles. Any two particles from the same cluster could be connected through a “continuous path”. By contrast, if there was no continuous path from one particle to another on an adjacent network, the two particles were considered not to belong to the same cluster. Here, if the distance between two particles was less than or equal to σ_{ij} , it was considered that the two particles were connected by continuous paths and belonged to the same cluster. If the number of particles in this aggregate was greater than N ($N = 10$ in this work), it was considered that the aggregation formed a cluster. The trajectories and cluster analysis were visualized using the OVITO software (Figure S19).

8) *Some of the letters in some figure (for example, Figure 4f) are too small to read. They should be enlarged for enhanced visibility.*

Response: We thank the reviewer for this good suggestion. We have revised the manuscript accordingly.

9) *The usage of general language including typos and grammatical errors need to be checked again.*

Response: We thank the reviewer for this good suggestion. We have revised the manuscript accordingly.

10) *Some of the digital figure are missing scale bars. They should be added to the pictures.*

Response: We thank the reviewer for this good suggestion. We have revised the manuscript accordingly.

Responses to the Reviewer #2

This paper reports a strategy of controlling nanofibril structure to create a powerful artificial muscle combining excellent mechanical and actuation performance. In the field of artificial muscles, it has always been a long-standing challenge to simultaneously realize high strength and work capacity, for example, increasing crosslinking density can improve mechanical strength but usually reduce actuation strain and work capacity. In this work, the reported method has successfully addressed such long-lasting challenge, representing a breakthrough in artificial muscle research. On the other hand, even though nanofibril structure has been reported in other polymer materials showing high strength and toughness, to the best of the reviewer's knowledge, these systems such as polyvinyl alcohol microfibers usually don't exhibit excellent actuation performance. Furthermore, in terms of actuation performance, it is very rare for soft actuators to simultaneously show such high work capacity, actuation stroke, actuation stress and work capacity. Based on these reasons, I would like to recommend the publication of this manuscript, provided that the following comments can be addressed.

Response: We thank the reviewer for these insightful comments, and we have revised

the manuscript according to these comments.

1. The key to this material design is the introduction of nanofibril structure, and the authors have found the significant mechanical properties enhancement as a result. However, the reviewer is still confused about the toughening mechanism even after finishing reading the manuscript. The authors should provide more explanations.

Response: Thank you very much for your valuable comments.

The nanofibrils, especially the nanodomain size of 2–6 nm achieved by self-assembly of the polypeptide chains in natural spider silk is considered as an important origin for its excellent mechanical strength and toughness^{20, 21}. Similarly, the mechanical properties of a fiber depend on the self-assembled structure^{17, 22, 23} and degree of alignment of the molecular chains²⁴⁻²⁶. The ordered arrangement of molecular chains can increase the fiber mechanical strength, and the self-assembled nanofibrils can inhibit crack propagation during deformation, thereby increasing energy dissipation²⁷⁻²⁹. (Page 2, Line 57–64)

The PAF_{4.75%} exhibited ultra-fine nanofibrils with average diameter of 17.7 nm. In this case, the crack propagation was effectively inhibited, and the nanofibrils were stretched out during mechanical stretching. For comparison, the PAF_{0.017%} and PAF_{96.5%} exhibited sharp cracking cross-section at fiber fracture. (Page 10, Line 297–301)

At appropriate dissociation degree (e.g. 4.75%), the polymer chains finely dissociated and are easily aligned with one another during the draw-spinning process. Then, during the water evaporation process, large amount of hydrogen bonding formed between the well-aligned polymer chains. The neighbor polymer chains got closely-packed and self-assembled to form nanofibrils. Increasing the alignment degree of the polymer chains in the nanofibril would allow the fiber withstand heavier loading stress along the fiber axis. Bundles of the polymer chains formed a lot of highly aligned nanofibrils. Inside the nanofibril strong interactions exist between the polymer chains, while the interaction between these nanofibrils are relatively weak. This can be confirmed by the fact that bundles of nanofibrils were pulled off from the longitudinal section of the fiber after fiber fracture. During mechanical stretching of a fiber, in the

case that there is a defect on the fiber surface, the nanofibrils in the defect would break to form a crack. The crack would be stopped and would not propagate to the neighbor nanofibrils, because of the relatively weak interaction and less entanglement between the polymer chains in the neighbor nanofibrils. While crack propagation and stress concentration in the defects of a common polymer material is considered as an important mechanical failure mechanism. Consequently, fracture of some nanofibrils of a fiber would not result in immediate breakage of the fiber, which would still withstand the loading stress because of the integrity of the remaining nanofibrils. In addition, the sliding, unfolding, elongation of nanofibrils in the fiber rather than crack all together during mechanical stretching would also help increasing the mechanical strength and toughness. Therefore, we observed increased mechanical strength, strain, and toughness for the fiber with obviously fine nanofibrils that exhibited increased polymer chain alignment. We have added this discussion in the context. (Page 11–12, Line 319–342)

Figure S20. (a) SEM images of the notched PAF α after breakage and (b) the fracture process of the notched PAF α during stretching.

Figure S21. Stress–strain curves of the original and notched PAF_α for different α values. The notch is 15 μm and the fiber diameter is 100 μm . The stretch rate is 5 mm min^{-1} .

2. *Is inserted twist structure stable? Because in most cases, the elasticity will result in complete untwist after removing the torsional tethering.*

Response: We thank the reviewer for this valuable comment. We then inserted twist into the PAF in the gel state and dried it to set the shape; this enables the realization of polymer chains with a spiral architecture by inducing a torsional stress in such a gel state⁴¹. As the twist was inserted in the fiber in the gel state, after twist insertion, both ends of the twisted fiber was tethered and the fiber was allowed to dry. Then, the inserted twist would be kept in the fiber because of formation of the hydrogen bonds between the polymer chains after water evaporation. We have added this discussion into the context. (Page 13, Line 376–382)

3. *How does the twist insertion affect the orientation degree? And how does it affect the mechanical properties?*

Response: We thank the reviewer for the valuable comments. We first inserted twist in a 95- μm -diameter $\text{PAF}_{4.75\%}$ that was prepared by photopolymerization in a capillary tube. With increase of the twist density, the orientation degree of nanofibrils and the mechanical strength and toughness increased to the maximum values, and then followed by decrease with further increase in the inserted twist density (Figure S25). As the twist density increased from 0 to 1.0 turns mm^{-1} , the orientation degree increased

from 86% to the maximum value of 88.7%, the breaking strength increased from 0.42 GPa to the maximum value of 0.50 GPa, and the toughness increased from 336 MJ m⁻³ to the maximum value of 498 MJ m⁻³; the orientation degree, breaking strength and toughness decreased with further increase in the twist density. We also inserted twist in a drawn-spun PAF_{1.1%} and investigated its mechanical properties. With increase in the twist density, the alignment degree of nanofibrils and mechanical strength monotonically decreased, the breaking strain and toughness monotonically increased (Figure S26). This would result from the fact that the molecular chains are already well-aligned in the drawn-spun PAF_{1.1%}, and twist insertion resulted in spiral configuration of the polymer chains. As the twist density increased from 0 to 16 turns mm⁻¹, the fracture strain of the PAF_{1.1%} increased monotonically from 18.8% to 35.8%, and the breaking stress decreased monotonically from 1.83 to 1.21 GPa, with the maximum toughness of 339 MJ m⁻³ obtained at the twist density of 12 turns mm⁻¹ (Figure 2d). We added this discussion in the context. (Page 13–14, Line 382–400)

Figure S25. (a) 2D SAXS patterns, (b) alignment degree of nanofibrils, (c) stress–strain curves of PAF_{4.75%} for different twist densities. The inset in (b) shows the azimuthal-integrated intensity distribution curves obtained from 2D SAXS patterns of PAF_{4.75%} with different twist densities. The stretch rate in the mechanical tests was 100 mm min⁻¹.

Figure S26. (a) 2D SAXS patterns, (b) alignment degree of nanofibrils, (c) stress–strain curves of drawn-spun PAF_{1.1%} for different twist densities. The inset in (b) shows the azimuthal-integrated intensity distribution curves obtained from 2D SAXS patterns and the inset in (c) shows metallographic microscopy images of drawn-spun PAF_{1.1%} with different twist densities. The fiber diameter for (a) was 30 μm . The fiber diameter for (c) was 5 μm . The stretch rate in the mechanical tests was 500 mm min^{-1} .

4. I suggest the authors to compare the actuation performance of twisted fiber to that of aligned fiber, which should be important for designing actuators with improved performance.

Response: We thank the reviewer for this good suggestion. We inserted twist in the 95- μm -diameter PAF_{4.75%} that was prepared by photopolymerization in a capillary tube to different twist densities, and investigated the fiber’s actuation performance at an actuation temperature of 100°C under different loading stresses (Figure S31). As the twist density increased from 0 to 1.0 turns mm^{-1} for the PAF_{4.75%} under the loading stress of 3.53 MPa, the actuation stroke increased from 40.9% to the maximum value of 58.5%, and the work capacity increased from 1.33 to the maximum value of 1.68 J g^{-1} ; with further increase of the twist density, the actuation stroke and the work capacity decreased. This showed the same dependence of fiber mechanical strength on the twist density. We added the discussion in the context. (Page 16, Line 460–468)

Figure S31. (a) Metallographic microscopy images of the PAF_{4.75%} fiber with different twist densities. (b) Actuation stroke and (c) work capacity of the PAF_{4.75%} as a function of isobarically loaded stress for different twist densities. (d) The maximum work capacity of the PAF_{4.75%} as a function of twist densities. The actuation temperature was 100 °C.

5. The fiber is actuated through shape memory mechanism, therefore, I think the authors should compare the mechanical properties and actuation performance to other thermoresponsive shape memory polymers, such as *Nat. Nanotechnol.* 2022, 17, 1198; and *ACS Central Science* 7, no. 10 (2021): 1657-1667. As the excellent mechanical and actuation performance have also been reported in both examples.

Response: We thank the reviewer for this valuable comment. In the revised manuscript, we added experiments to optimize the actuation performances of the PAF_α. A 5-μm-diameter drawn-spun PAF_{1.1%} exhibited an actuation stress of 65 MPa as the temperature increased from 25°C to 120°C when the fiber was the both-end tethered on the mechanical tester. The combination of breaking strength (1.8 GPa) and actuation stress (65 MPa) of the PAF_α is among the best of the hydrogel materials and shape memory polymer materials, such as PAAM/PAA/cellulose nanocrystals fibers (0.065

GPa for breaking strength and 0.24 MPa for actuation stress), PAA/cotton fibers (0.15 GPa for breaking strength and 0.03 MPa for actuation stress), and PAA/carbon nanotube fibers (0.2 GPa for breaking strength and 0.01MPa for actuation stress), COCe-PE bimorph fiber (0.0258 GPa for breaking strength and 5.0 MPa for actuation stress), block polymer (0.146 GPa for breaking strength and 5.5 MPa for actuation stress), LCE-Graphene fiber (0.09 GPa for breaking strength and 1.23 MPa for actuation stress) (Figure 4e, 4f, 4g and Figure 1f, Table S10 and Table S11). We have added this discussion in the context of the manuscript. (Page 17, Line 488–497)

Figure 4. (e) Actuation stroke and work capacity of a 5- μm drawn-spun PAF_{1.1%} as a function of isobarically loaded stress. The actuation temperature was 100°C. (f) Contractile actuation stress of the drawn-spun PAF_{1.1%} as a function of time by heating from 25°C to 120°C by employing two-parallel heating plates measured on the mechanical tester. The drawn-spun PAF_{1.1%} was both-end tethered on the holder of the mechanical tester to obtain the actuation stress.

Figure 1. (f) Comparison of the actuation stress and mechanical strength of the drawn-spun PAF_{1.1%} prepared in this work with those of typical thermoresponsive shape memory polymer artificial muscles reported in the literature.

Figure 4. (g) Comparison of the work capacity and actuation stroke of the drawn-spun PAF_{1.1%} prepared in this work with those of typical artificial hydrogel muscles reported in the literature.

Table S11. Comparison of the actuation performance of typical thermally supercontraction artificial muscles in literature.

Materials	Break stress (MPa)	Actuation stroke (%)	Actuation stress (MPa)	Work capacity (J g⁻¹)	Ref.
Mammalian skeletal muscles	0.47	40	0.35	0.039	47
Semicrystallie polymer fibers	-	49	1.0	2.48	48
COCe-PE bimorph fiber	25.8	47.7	5.0	7.42	49
Block polymer	146.2	30.8	5.5	0.506	47
Supramolecular Nanostructures ploymer	70	300	13.1	-	50
LCE-Graphene	90	45	1.23	0.65	51
LCE	80	274	1.3	0.73	52
	2.53	40	0.35	1.83	53
	3.2	296	0.1	0.296	54
PU	25	24	1.0	0.708	55
	12	65	0.002	0.717	56
	12	-	0.023	0.0725	57
PU/LCE/AU	12	66	0.001	0.87	58
This work	530-1800	11-56.3	6.22-65	1.58-2.77	

Table S10 Comparison of the work capacity and actuation stroke of typical artificial muscles in literature.

artificial muscles	Break stress (MPa)	Actuation stress (MPa)	Actuation stroke (%)	Work capacity (J g⁻¹)	Ref.
PAA-Polyester fiber	-	1.0	70	1.0	37
PAA/Ca(CH ₃ COO) ₂ fiber	14	-	52.7	0.0452	38
PAA/cotton fiber	150	0.03	61	0.0061	39
PAA/CNT fiber	200	0.01	56	0.0091	40
PAAM/PAA/TCNC fiber	65	0.24	36	0.0424	41
PAM/PAA/PSMA/Q-TCNC fiber	32.6	3.0	75	0.21	42
Nylon/BA-PA fiber	0.73	0.08	1.4	0.13	43
Linen yarn	-	-	10	0.11	44
Human hair	200	0.25	20	0.006	46
This work	530-1800	6.22-65	11-56.3	1.58-2.77	

6. *Is it possible to realize two-way reversible actuation? It will be more desirable for the real-world applications.*

Response: We thank the reviewer for this valuable comment. To achieve two-way reversible actuation, twist was inserted in the PAF_α to form self-coil or mandrel-coil architectures (Figure S34–35). Briefly, taking the PAF_{4.75%} as an example, the PAF_{4.75%} was polymerized in a capillary tube, prestrained to 1200%, and air-dried for 15 min at RH 10% to obtain a 95-μm-diameter fiber. To prepare a self-coiled artificial muscle, we insert twist (4.5 turns mm⁻¹) into a 95-μm-diameter PAF_α until fiber coiling under a constant load of 3.3 MPa, followed by cross-linking by employing Zr⁴⁺ to set the coiled shape to obtain a self-supporting artificial muscle with spring index of 1.17. We then measured the actuation stress of the self-coiled PAF_{4.75%} by tethering the both-ends of the sample on the mechanical tester. By heating the self-coiled PAF_{4.75%} artificial muscle from 25°C to 120°C, an actuation stress of 9.0 MPa was obtained, which was repeated for several heating-cooling cycles (Figure S34).

In addition, we also prepared mandrel-coiled artificial muscles by wrapping the twisted PAF_{4.75%} (2.0 turns mm⁻¹) around a mandrel, followed by cross-linking via Zr⁴⁺ to set the coiled shape (Figure S35). The actuation performances were investigated for

the homochiral PAF_{4.75%} artificial muscles with different twist densities and spring index. The mandrel-coiled PAF_{4.75%} artificial muscles exhibited high reversibility during repeated heating and cooling cycles. We measured the dependence of actuation stroke of the mandrel-coiled self-supporting PAF_{4.75%} artificial muscle with different twist densities and spring indexes without a load. The actuation stroke increased from 25% to 65% as the spring index increased from 10 to 50 for the mandrel-coiled PAF_{4.75%} artificial muscle with twist density of 2.0 turns mm⁻¹. The actuation stroke increased from 20% to 65% as the twist density increased from 1.0 to 3.0 turns mm⁻¹ for the mandrel-coiled PAF_{4.75%} artificial muscle with spring index of 30. We have added this discussion in the context of the manuscript. (Page 18–19, Line 526–549)

Figure S34. Representative curve of the contractile stress of self-coiled PAF_{4.75%} artificial muscle in response to rapid increase in temperature from 25°C to 120°C via two-parallel heating plates. The twist density was 4.5 turns mm⁻¹, the fiber diameter was 95-μm, the spring index was 1.17, and the inserted twist was preserved by cross-linking using Zr⁴⁺ to obtain a self-supporting artificial muscle. Scale bar: 100 μm.

Figure S35. (a) Schematic illustration of the fabrication of the single homochiral hydrogel fiber artificial muscles. (b) Three cycle of the actuation stroke as a function of time for the homochiral hydrogel fiber coils in response to the temperature changes between 60°C and 25°C. The twist density is 2.0 turns mm⁻¹ and the spring index is 30. (c) Actuator performance for the coiled homochiral hydrogel fiber muscles as a function of (c) spring index and (d) twist density.

7. How about the actuation performance at lower actuation temperature?

Response: We thank the reviewer for the valuable suggestion. We investigated the actuation stroke of the 30- μ m-diameter drawn-spun PAF_{1.1%} at different temperatures between 40°C to 100°C without loading a mass. The drawn-spun PAF_{1.1%} contracted by 38.3% in 8.0 s at 40°C and reached a plateau. The response time decreased from 8.0 s to 2.0 s as the actuation temperature increased from 40°C to 100°C, corresponding to a maximum actuation speed increasing from 12.2% s⁻¹ to 173.8% s⁻¹. We have added the experimental results and discussions in the context. (Page 15, Line 440–445)

Figure S32. (a) The actuation strain of drawn-spun PAF_{1.1%} as a function of time at different actuation temperatures. (b) The maximum actuation speed of the drawn-spun PAF_{1.1%} as a function of actuation temperature. The diameter of PAF_{1.1%} is 30 μm .

Responses to the Reviewer #3:

This work prepared a polyelectrolyte artificial spider silk with thermal-induced supercontraction ability. It is indicated that the high-performance manmade fiber can be used as artificial muscle, and the rope made from silk can pull a car. I am sure that many existing ropes can withstand the car pulling to some extent. However, this work is not presented logically and some fundamental things are wrong and the data for the presented figures are not complete. Thus, I don't think it is suitable for publication in Nature Communications. Here are some concerns and questions:

Response: We thank the reviewer for these valuable comments to improve the manuscript. In the revised manuscript, we re-organized the figures and context to present the manuscript more logically, supplemented the data for the presented figures, add more experiments and discussions to increase the performance of the fiber materials.

1. What's the definition of artificial spider silk? This fiber does not share a similar structure to the natural spider silk.

Response: We have added contents in the introduction part and added a schematic to show the similarity of strong and tough mechanical properties and hierarchical structure of the natural spider silk and artificial spider silk. The revised introduction and Figure 1 is as follows:

Acting over evolutionary time scales, nature has generated astonishing nanostructures via the self-assembly of bio-macromolecules, yielding diverse tough materials with extraordinary mechanical properties¹⁻⁴. Spider silk exhibits an excellent combination of mechanical strength and toughness, which originates from the hierarchical structure assembled by the protein peptide chains. The hierarchical structures of spider silk include nanofibrils with spiral structure formed by highly oriented peptide chains, β -sheet crystallites serving as physical cross-linking sites, hydrogen bonding to dissipate energy, and sheath-core architecture with rigid sheath and soft core⁵. By mimicking some of these structural characteristics, great achievements have been realized in preparation of artificial spider silks with high strength and toughness by employing spidroin proteins and peptides^{6, 7}, carbon nanotube/polymer composite fibers^{8, 9}, and hydrogel fibers^{10, 11}.

Artificial spider silks based on non-peptide synthetic polymers are prepared by mimicking the hierarchical structure of the spider silk, including sheath-core^{12, 13}, spiral alignment by inserting twist^{14, 15}, cross-linking^{11, 16}, and combination of the above structures^{10, 17}. Among these methods, draw-spinning of the polymer hydrogel fibers are proven to be an effective way to prepare spider-silk-like mechanical properties, with hierarchical structures including sheath-core and spiral alignment¹⁰, internal cross-linking and hydrogen bonding¹⁷, buckled sheath¹⁸, and adhesiveness¹⁹. Until now, to modulate the assembly of molecular chains in the hierarchical structure of the polymer artificial spider silk is still an on-going challenge to further improve the fiber mechanical properties.

The nanofibrils, especially the nanodomain size of 2–6 nm achieved by self-assembly of the polypeptide chains in natural spider silk is considered as an important origin for its excellent mechanical strength and toughness^{20, 21}. Similarly, the mechanical properties of a fiber depend on the self-assembled structure^{17, 22, 23} and degree of alignment of the molecular chains²⁴⁻²⁶. The ordered arrangement of molecular chains can increase the fiber mechanical strength, and the self-assembled nanofibrils can inhibit crack propagation during deformation, thereby increasing energy dissipation²⁷⁻²⁹. Therefore, being able to precisely control the self-assembly of molecular chains in nanostructures is key to improving the fiber mechanical strength and toughness³⁰. Although assembly of polymer chains into nanofibrils were observed in polymer materials^{31, 32}, it is still a challenge to regulate the self-assembly of

nanofibrils in the hierarchical structure of the polymer artificial spider silk to achieve excellent mechanical properties.

In this work, a robust artificial spider silk was prepared by establishing superfine nanofibrils by optimizing the molecular chain flexibility to modify the self-assembling process. An increased degree of molecular chain alignment of polyelectrolyte was achieved through an increased dissociation degree during solvent evaporation under external stress, resulting in excellent tunable mechanical properties. For example, the polyacrylic acid fiber (PAF) exhibited a combination of mechanical strength and toughness ranging from 1.83 GPa and 238 MJ m⁻³ to 0.53 GPa and 700 MJ m⁻³, respectively. A nanodomain size of 5.2 nm were observed in the self-assembly structure, approaching that of natural spider silk²⁰, which is considered as an important origin for improving the mechanical strength and toughness.

A unique thermally driven supercontraction behavior was observed for the PAF-based artificial spider silk. Interestingly, the establishment of superfine nanofibrils provided the obtained PAF-based artificial spider silk highly increased actuation properties. The PAF exhibited the maximum actuation stress of 65 MPa and the maximum work capacity of 2.77 J g⁻¹ for the optimized dissociation degree. Further twist insertion followed by cross-linking produced coiled artificial muscles with reversible actuation. The current work provides a new design strategy for high-performance, smart fibers for use in soft robotics, flexible electronics and intelligent devices. (Page 2-3, Line 35-88)

Figure 1. Preparation and characterization of the PAF $_{\alpha}$ -based artificial spider silk. **(a)** Schematic of the spinning of spider silk and the PAF $_{\alpha}$ artificial spider silk. The modulation of the nanofibrils of the PAF $_{\alpha}$ achieved by tuning the polymer chain flexibility, which in turns is obtained by changing the dissociation degree. **(b)** and **(c)** The coarse-grained MD simulations show that the polymer chain clusters transform from a multi-branched shape to highly dissociated as α increases from 0% to 25%. **(d)** AFM image of the longitudinal-sectional nanofibrils of the PAF $_{0.17\%}$. **(e)** Comparison of the breaking stress and toughness of the artificial spider silk fibers in this work with those of typical robust fiber materials reported in the literature. **(f)** Comparison of the actuation stress and mechanical strength of the drawn-spun PAF $_{1.1\%}$ prepared in this work with those of typical thermoresponsive shape memory polymer artificial muscles reported in the literature.

2. It is extremely misleading in Figure 1, where a 4-ply-rope with 3000 fibers and twists is compared with a single spider silk fiber in literature. The images and charts for comparison there are not fair and not logical. Their so called high-performance manmade fiber can be used as artificial muscle, and the rope made from silk can pull a car. I am sure that many existing ropes can withstand the car pulling to some extent. Thus, this material is not that magical as they claimed.

Response: Thank you very much for this suggestion. We have removed the demo of drawing a car by employing the artificial spider silk rope from Figure 1, added the schematic of the hierarchical structure of the natural spider silk and artificial spider silk, added the images showing the nanofibril structure, and added the comparison of PAF α artificial muscle with literature reported results.

3. How the authors test the mechanical properties (for example, on a single silk, ply silk, two-ply, four-ply, or rope)? This should be indicated in the manuscript otherwise the data are not complete. Besides, there are also many basic charts are not presented such as stress-strain curves of tensile testing, thus collected data can not be reasoned.

Response: Thank you very much for this comment. We added the experimental details for the mechanical characterization of the PAF fiber, as follows. If not specified, the stress–strain test was carried out on a single PAF on an Instron mechanical tester. The PAFs for mechanical test were taped onto the paper frames with a gauge length of 6 mm. The environmental temperature was 25°C, and the relative humidity was 10%. The final mechanical properties were the average values of 5 independent tests. The stress–strain curves for Figure 2b, 2c, 2d and 2e are shown in Figure S9, S24, S26 and S21, respectively. We have added the above information into the supplementary information (Page 13, Page 19, Page 21, Page 22).

The corresponding stress–strain curves of tensile testing are as follows.

Figure S9. Stress–strain curves of the PAF_α for different α values. The fiber diameter was 80 μm. The stretch rate in the mechanical tests was 200 mm min⁻¹. (Corresponding to Figure 2b)

Figure S24. Stress–strain curves of the draw-spun PAF_α for different α values. The fiber diameter was 5 μm. The stretch rate in the mechanical tests was 20 mm min⁻¹. (Corresponding to Figure 2c)

Figure S27. (a) Stress–strain curves and (b) breaking stress, breaking strain, and toughness of the draw-spun PAF_{1.1%} for stretching rate. The fiber diameter was 5 μm .

Figure S26. (c) Stress–strain curves of the draw-spun PAF_{1.1%} for different twist densities. The fiber diameter was 5 μm . The stretch rate in the mechanical tests was 500 mm min^{-1} . (Corresponding to Figure 2d)

Figure S21. Stress–strain curves of the original and notched PAF_α for different α values. The notch is 15 μm and the fiber diameter is 100 μm . The stretch rate is 5 mm min^{-1} . (Corresponding to Figure 2e)

4. How to calculate work capacity? How to make a notch on the fiber? How to get the actuation stress? The author mentioned inserting twist in the fiber. What's the twisting parameter? The Method section should be elaborated in detail.

Response: We added these experimental descriptions in Method section, as follows. The work capacity (J g^{-1}) can then be calculated directly:

$$W = \frac{m}{m_0} \mathbf{g}l,$$

where m is the loading mass, m_0 is the fiber mass, l is the displacement of the loading mass during actuation, and \mathbf{g} is the acceleration constant of gravity.

To make a notch on the fiber, a sharp blade was employed to make a 15- μm -deep single-edge cut in the radial direction of a 100- μm -diameter PAF_α that has been dried for 3 h in 60°C. Then, the fiber with notch was then subjected to mechanical test at a stretch rate of 5 mm min^{-1} .

The actuation properties were characterized by two methods in the revised manuscript, as follows. In the method 1, the actuation of the PAF_α was characterized by lifting a load by heating the fiber to an elevated temperature. The PAF_α with radius of r and length l_0 was loaded with a mass of m , and then the PAF_α was heated to an elevated temperature. The PAF_α contracted to length of l_1 . The actuation stress was

calculated as $mg/\pi r^2$, and the actuation strain was calculated as $(l_0 - l_1)/l_0 \times 100\%$.

In the method 2 (added in the revised manuscript), the actuation of the PAF_α was characterized by directly measuring the contractile force on the mechanical tester by heating the fiber to an elevated temperature. The PAF_α was both-end tethered on the clamps of the mechanical tester. Then, the PAF_α was heated to an elevated temperature to cause contractile force (F), which is measured by the mechanical tester. The actuation stress was calculated as $F/\pi r^2$.

For twist insertion, the wet as-obtained PAF_α before drying was employed. One end of a 25-mm-long PAF_α was vertically connected to an 80-step servomotor, and the other end was isobarically loaded (constant load 3.53 MPa) with a stainless-steel ring. An iron rod was inserted into the stainless-steel ring to prevent the PAF_α from rotating during twist insertion. Then, twist was inserted by rotation of the servomotor. Different twist densities were inserted in the fibers. After twist insertion, the PAF_α was dried for 3 h in 60°C for mechanical properties tests.

We have added the above experimental details in the supplementary information. (Page 3–4, Line 84–110)

5. The supercontraction presented in the article is a one-time response. How to get a continuous cycle of the tensile stroke in Figure 4e. The actual recycling data there may be very rough, but the presentation seems to be very effective. This is also a misleading factor.

Response: We added the experimental details and a schematic to show the cycling process of the supercontraction of PAF_α, as follows.

A 95-μm-diameter PAF_α loaded with 1.45 MPa mass at room temperature (25°C) was heated to 100°C to allow supercontraction to the maximum actuation stroke. Then, the PAF_α was cooled to room temperature, and the fiber would keep at this contracted length. The PAF_α was then exposed to ultrasonically generated water fog (95% relative humidity) and re-stretched to the initial length, followed by air-drying for 15 min to set the length. The above cycling process was repeated for 50 times. The schematic of the above process was shown in the inset of Figure 4d. (Page 16, Line 469–475)

Figure 4. (d) Actuation stroke of the PAF_{4.75%} under an isobaric load of 1.45 MPa as a function of the number of cycles. The inset shows the schematic of the process.

In the revised manuscript, we added experiments of two-way reversible actuation, by inserting twist in the PAF _{α} to form self-coil or mandrel-coil architectures. Good actuation reversibility was achieved during repeated heating and cooling cycles. (Figure S34–35). Briefly, taking the PAF_{4.75%} as an example, the PAF_{4.75%} was polymerized in a capillary tube, prestrained to 1200%, and air-dried for 15 min at RH 10% to obtain a 95- μm -diameter fiber. To prepare a self-coiled artificial muscle, we insert twist (4.5 turns mm^{-1}) into a 95- μm -diameter PAF _{α} until fiber coiling under a constant load of 3.3 MPa, followed by cross-linking by employing Zr^{4+} to set the coiled shape to obtain a self-supporting artificial muscle with spring index of 1.17. We then measured the actuation stress of the self-coiled PAF_{4.75%} by tethering the both-ends of the sample on the mechanical tester. By heating the self-coiled PAF_{4.75%} artificial muscle from 25°C to 120°C, an actuation stress of 9.0 MPa was obtained, which was repeated for several heating-cooling cycles (Figure S34).

In addition, we also prepared mandrel-coiled artificial muscles by wrapping the twisted PAF_{4.75%} (2.0 turns mm^{-1}) around a mandrel, followed by cross-linking via Zr^{4+} to set the coiled shape (Figure S35). The actuation performances were investigated for the homochiral PAF_{4.75%} artificial muscles with different twist densities and spring index. The mandrel-coiled PAF_{4.75%} artificial muscles exhibited high reversibility during repeated heating and cooling cycles. We measured the dependence of actuation

stroke of the mandrel-coiled self-supporting PAF_{4.75%} artificial muscle with different twist densities and spring indexes without a load. The actuation stroke increased from 25% to 65% as the spring index increased from 10 to 50 for the mandrel-coiled PAF_{4.75%} artificial muscle with twist density of 2.0 turns mm⁻¹. The actuation stroke increased from 20% to 65% as the twist density increased from 1.0 to 3.0 turns mm⁻¹ for the mandrel-coiled PAF_{4.75%} artificial muscle with spring index of 30. We have added this discussion in the context of the manuscript (Page 18–19, Line 526–549).

Figure S34. Representative curve of the contractile stress of self-coiled PAF_{4.75%} artificial muscle in response to rapid increase in temperature from 25°C to 120°C via two-parallel heating plates. The twist density was 4.5 turns mm⁻¹, the fiber diameter was 95-μm, the spring index was 1.17, and the inserted twist was preserved by cross-linking using Zr⁴⁺ to obtain a self-supporting artificial muscle. Scale bar: 100 μm.

Figure S35. (a) Schematic illustration of the fabrication of the single homochiral hydrogel fiber artificial muscles. (b) Three cycle of the actuation stroke as a function of time for the homochiral hydrogel fiber coils in response to the temperature changes between 60°C and 25°C. The twist density is 2.0 turns mm^{-1} and the spring index is 30. (c) Actuator performance for the coiled homochiral hydrogel fiber muscles as a function of (c) spring index and (d) twist density.

6. The author explained that the high breaking strength and toughness of the fibers are achieved by introducing ions to control the dissociation of polymer chains and evaporation induced self-assembly under external stress. This part is investigated in detail. However, I want to question how the dissociation degree (α) influences the supercontraction behavior. What is the mechanism for the supercontraction. Contraction upon heating is common in many synthetic polymers and how to distinguish the supercontraction in this fiber from the others? It is mentioned on Page 12 that “Until now the thermally driven supercontraction of hydrogel fibers was rarely reported.” And how about other synthetic fibers?

Response: Thank you very much for these questions! We added experiments and

discussions to answer these questions.

We investigated the supercontraction performance of PAF_α with different α values, and also correlated with alignment degree of molecular chains and nanofibrils. As α increased from 0.017% to 4.75%, the actuation strain increased from 8.9% to the maximum value of 65%, and the work capacity increased from 0.2 J g^{-1} to the maximum value of 1.5 J g^{-1} , by heating the PAF_α loaded with 2.7 MPa from 25°C to 100°C ; at the same time, the orientation degree of molecular chains (f_m) increased from 36.3% to the maximum value of 56.5%, and the orientation degree of nanofibrils (f_n) increased from 90.3% to the maximum value of 93.3%; as α further increased to 21.9%, we observed decrease of the actuation strain, work capacity, f_m , and f_n (Figure 4c). (Page 15, Line 445–454)

Figure 4. (c) The dependence of the actuation strain, work capacity, the orientation degree of molecular chains (f_m), and the orientation degree of nanofibrils (f_n) of the PAF_α on α values. The PAF_α was loaded with 2.7 MPa by heating from 25°C to 100°C for actuation.

The heating-induced supercontraction of PAF_α should be ascribed to the morphology change of the highly oriented molecular chains (exhibiting low entropy state) to the low oriented molecular chains (exhibiting high entropy state). This process is similar to the shape memory behavior, and the proposed mechanism is as follows. The as-prepared PAF_α after polymerization contains a large amount of water molecules,

and the hydrogen bonding between the polymer chains was disrupted by the water molecules. Therefore, the fiber exhibited high elasticity, and the molecule chains exhibited low orientation and high entropy. Then, the as-prepared PAF_α was pre-stretched to an elongated length and air dried. The molecular chains after pre-stretch exhibited high orientation and low entropy. After water evaporation, the hydrogen bonding between the polymer chains was re-constructed, and the PAF_α kept at this elongated length. By heating, the hydrogen bonding between the polymer chains was disrupted at an elevated temperature, and the polymer chains spontaneously changed to the morphology with low orientation degree (exhibiting high entropy). Figure S33 shows the 2D WAXS patterns before and after thermally driven supercontraction, which shows decreased anisotropy and alignment degree of molecular chains after thermally driven supercontraction.

Different from the common shape-memory polymer materials, after thermally driven supercontraction, the hydrogel-based PAF_α can be easily re-stretched to the elongated length after wetting, followed by air-dry to set the shape for the next round of supercontraction by heating. Such a process is a synergistic combination of entropy-driven morphology change of polymer chains and the breaking and re-formation of hydrogen bonding between the polymer chains of the PAF_α. Negligible decay of the actuation performance of the PAF_α was observed for 50 cycles of thermally driven supercontraction (Figure 4d). Moreover, because the PAF_α exhibited spider-silk like hierarchical structure with tunable nanofibrils of the polymer chains, we further optimized the supercontraction capacity of the PAF_α, which exhibited excellent actuation performance with maximum actuation stress up to 65 MPa (Figure 4f). (Page 17–18, Line 498–525)

Figure S33. Azimuthal-integrated intensity distribution curve from 2D WAXS of PAF_{4.745%} before and after actuation.

Figure 4. (f) Contractile actuation stress of the drawn-spun PAF_{1.1%} as a function of time by heating 25°C to 120°C by employing two-parallel heating plates measured on the mechanical tester. The drawn-spun PAF_{1.1%} was both-end tethered on the holder of the mechanical tester to obtain the actuation stress.

7. The actuation stress is 2.7 MPa only, which is extremely low, how could the authors say this is powerful muscle? The actuation stress of natural spider silk can reach 80

MPa. Data in Figure 4 seems to conflict with the mechanical properties mentioned in the other part. Figure 4a shows a silk lifting a load. Is this a single PAF, or PAF with twist, or plied PAF? For example, Figure 4a shows a silk can lift a 2g load with a stroke of 62%. However, when the load reaches 8 MPa, there is no tensile stroke (Figure 4c), which seems the actuation force is very low. It does not match their claim and data for high strength of the fiber.

Response: We added experiments and discussions to answer these questions. We obtained optimized actuation stress by employing the drawn-spun PAF_α with different diameters, by directly measuring the actuation stress on the mechanical tester. This was realized by holding the both ends of the PAF_α on the clamp of the mechanical tester and heating the fiber to an elevated temperature via two-parallel heating plates. The drawn-spun $\text{PAF}_{1.1\%}$ with diameter of 75, 50, 30 and 5 μm were investigated. As the diameter of the $\text{PAF}_{1.1\%}$ decreased from 75 to 5 μm , the actuation stress increased from 13.6 to 65 MPa, which is in the same level of the actuation stress of the natural spider silk (~ 80 MPa)⁴⁸. We further investigated the load-lifting capacity of the 5 μm -diameter drawn-spun $\text{PAF}_{1.1\%}$. The 5- μm -diameter drawn-spun $\text{PAF}_{1.1\%}$ can lift a 9.33 MPa load by an actuation stroke of 37.5%, corresponding to an actuation work capacity of 2.77 J g^{-1} ; it also can lift a load of 18.6 MPa by an actuation stroke of 11%, corresponding to an actuation work capacity of 1.58 J g^{-1} (Figure 4e and f).

The combination of breaking strength (1.8 GPa) and actuation stress (65 MPa) of the PAF_α is among the best of the hydrogel materials and shape memory polymer materials, such as PAAM/PAA/cellulose nanocrystals fibers (0.065 GPa for breaking strength and 0.24 MPa for actuation stress), PAA/cotton fibers (0.15 GPa for breaking strength and 0.03 MPa for actuation stress), and PAA/carbon nanotube fibers (0.2 GPa for breaking strength and 0.01MPa for actuation stress), COCe-PE bimorph fiber (0.0258 GPa for breaking strength and 5 MPa for actuation stress), block polymer (0.146 GPa for breaking strength and 5.5 MPa for actuation stress), LCE-Graphene fiber (0.09 GPa for breaking strength and 1.23 MPa for actuation stress) (Figure 4g and Figure 1f, Table S10 and Table S11). (Page 16–17, Line 476–497)

Figure 4. (e) Actuation stroke and work capacity of a 5- μm drawn-spun PAF_{1.1%} as a function of isobarically loaded stress. The actuation temperature was 100°C. (f) Contractile actuation stress of the drawn-spun PAF_{1.1%} as a function of time by heating from 25°C to 120°C by employing two-parallel heating plates measured on the mechanical tester. The drawn-spun PAF_{1.1%} was both-end tethered on the holder of the mechanical tester to obtain the actuation stress.

Figure 4. (g) Comparison of the work capacity and actuation stroke of the drawn-spun PAF_{1.1%} prepared in this work with those of typical artificial hydrogel muscles reported in the literature.

Figure 1. (f) Comparison of the work capacity and mechanical strength of the drawn-spun PAF_{1.1%} prepared in this work with those of typical thermoresponsive shape memory polymer artificial muscles reported in the literature.

Table S10. Comparison of the work capacity and tensile stroke of typical artificial muscles in literature.

artificial muscles	Break stress (MPa)	Actuation stress (MPa)	Actuation stroke (%)	Work capacity (J g ⁻¹)	Ref.
PAA-Polyester fiber	-	1.0	70	1.0	37
PAA/Ca(CH ₃ COO) ₂ fiber	14	-	52.7	0.0452	38
PAA/cotton fiber	150	0.03	61	0.0061	39
PAA/CNT fiber	200	0.01	56	0.0091	40
PAAM/PAA/TCNC fiber	65	0.24	36	0.0424	41
PAM/PAA/PSMA/Q-TCNC fiber	32.6	3.0	75	0.21	42
Nylon/BA-PA fiber	0.73	0.08	1.4	0.13	43
Linen yarn	-	-	10	0.11	44
Human hair	200	0.25	20	0.006	46
This work	530-1800	6.22-65	11-56.3	1.58-2.77	

Table S11. Comparison of the actuation performance of typical thermally supercontraction artificial muscles in literature.

Materials	Break stress (MPa)	Actuation stroke (%)	Actuation stress (MPa)	Work capacity (J g⁻¹)	Ref.
Mammalian skeletal muscles	0.47	40	0.35	0.039	47
Semicrystallie polymer fibers	-	49	1.0	2.48	48
COCe-PE bimorph fiber	25.8	47.7	5.0	7.42	49
Block polymer	146.2	30.8	5.5	0.506	47
Supramolecular Nanostructures ploymer	70	300	13.1	-	50
LCE-Graphene	90	45	1.23	0.65	51
LCE	80	274	1.3	0.73	52
	2.53	40	0.35	1.83	53
	3.2	296	0.1	0.296	54
PU	25	24	1.0	0.708	55
	12	65	0.002	0.717	56
	12	-	0.023	0.0725	57
PU/LCE/AU	12	66	0.001	0.87	58
This work	530-1800	11-56.3	6.22-65	1.58-2.77	

8. *The authors present so much data in the manuscript, and these data are short of backing evidences for understanding.*

Response: Thank you very much! We have substantially revised this manuscript, re-written the introduction part, added additional experiments, discussion, and additional data for backing evidence, re-organize the figures. Now this manuscript has been improved for better understanding of the readers.

REVIEWERS' COMMENTS

Reviewer #1 (Remarks to the Author):

The authors responded well to the comments. This should be ready for publication now.

Reviewer #2 (Remarks to the Author):

The authors have generally addressed the issues raised by the reviewers in the revised manuscript and in the response letter. Several minor points need revisions:

1. In the revised manuscript, the authors commented that the combination of breaking strength (1.8 GPa) and actuation stress (65 MPa) of PAF is among the best of the hydrogel materials. However, I don't think the sample showing such high strength and actuation stress is hydrogel. Because the sample has been dried in an oven at 60 °C before measuring the mechanical and actuation performance. Please correct it.

2. In the revised manuscript, the authors observed the reduced work capacity with further increase of the twist density. An explanation is needed here as it has been widely reported that the twist insertion can improve actuation performance.

Reviewer #3 (Remarks to the Author):

The authors have addressed most of the comments well. However, I still have one more comment for the authors. Regarding the response to question 5, the authors have expressed the measuring of tensile stroke. The contraction is a one-time response and needs to go through a stretching-fixing process before the next stroke test. Therefore, I don't think this is a continuous cyclic response, and the presentation in Figure 4b is still misleading. The data should not be presented in a continuous line chart. I suggest presenting the tensile stroke data in individual dot format.

Reviewer #1 (Remarks to the Author):

The authors responded well to the comments. This should be ready for publication now.

Response: Thank you very much for your kind effort for improving this article.

Reviewer #2 (Remarks to the Author):

The authors have generally addressed the issues raised by the reviewers in the revised manuscript and in the response letter. Several minor points need revisions:

Response: Thank you very much for your kind effort for improving this article.

1. In the revised manuscript, the authors commented that the combination of breaking strength (1.8 GPa) and actuation stress (65 MPa) of PAF is among the best of the hydrogel materials. However, I don't think the sample showing such high strength and actuation stress is hydrogel. Because the sample has been dried in an oven at 60 °C before measuring the mechanical and actuation performance. Please correct it.

Response: We thank the reviewer for this comment, and we have corrected the description as follows:

Because PAF_α has been dried in an oven at 60°C before measuring the mechanical and actuation performance, it is not a hydrogel material. The combination of breaking strength (1.8 GPa) and actuation stress (65 MPa) of the PAF_α is among the best of the actuation materials including hydrogel materials and shape memory polymer materials, ...(Page 17, Line 490)

2. In the revised manuscript, the authors observed the reduced work capacity with further increase of the twist density. An explanation is needed here as it has been widely reported that the twist insertion can improve actuation performance.

Response: We thank the reviewer for this important comment. We have added discussion to explain this effect, as follows.

There are two effects of the polymer chain orientation by twist insertion of the PAF_α fiber in the gel state. The polymer chains exhibited a random coil morphology in the as prepared fiber. During twist insertion the polymer chains exhibited a spiral alignment under the torsional stress. The increased orientation of the polymer chains increased the mechanical stress and the actuation stress, while spiral architecture of the polymer chains resulted in a twist angle between the polymer chain orientation and the fiber axial direction, causing a decrement in the axial contribution of the mechanical stress and the actuation stress. Therefore, with increase of the inserted twist density, the work capacity of the PAF_α first increased to a maximum value and then decreased. (Page 16, Line 461)

Reviewer #3 (Remarks to the Author):

The authors have addressed most of the comments well. However, I still have one more comment for the authors. Regarding the response to question 5, the authors have expressed the measuring of tensile stroke. The contraction is a one-time response and needs to go through a stretching-fixing process before the next stroke test. Therefore, I don't think this is a continuous cyclic response, and the presentation in Figure 4b is still misleading. The data should not be presented in a continuous line chart. I suggest presenting the tensile stroke data in individual dot format.

Response: Thank you very much for your kind effort for improving this article. We have revised the data Figure 4d into individual dots, as follows.

Figure 4. (d) Actuation stroke of the PAF_{4.75%} under an isobaric load of 1.45 MPa as a function of the number of cycles. The inset shows the schematic of the process.